# A reactive neural network framework for water-loaded acidic zeolites

Andreas Erlebach [1] ✉, Martin Šípka[1,2], Indranil Saha[1], Petr Nachtigall[1,3], Christopher J. Heard [1] & Lukáš Grajciar [1] ✉

Under operating conditions, the dynamics of water and ions confined within protonic aluminosilicate zeolite micropores are responsible for many of their properties, including hydrothermal stability, acidity and catalytic activity. However, due to high computational cost, operando studies of acidic zeolites are currently rare and limited to specific cases and simplified models. In this work, we have developed a reactive neural network potential (NNP) attempting to cover the entire class of acidic zeolites, including the full range of experimentally relevant water concentrations and Si/Al ratios. This NNP has the potential to dramatically improve sampling, retaining the (meta)GGA DFT level accuracy, with the capacity for discovery of new chemistry, such as collective defect formation mechanisms at the zeolite surface. Furthermore, we exemplify how the NNP can be used as a basis for further extensions/ improvements which include data-efficient adoption of higher-level (hybrid) references via Δ-learning and the acceleration of rare event sampling via automatic construction of collective variables. These developments represent a significant step towards accurate simulations of realistic catalysts under operando conditions.

Zeolites are a class of microporous aluminosilicates with tremendous structural and chemical diversity, which originates from the myriad stable three-dimensional arrangements of covalently connected silica/ alumina tetrahedra. This makes zeolites a versatile material class with applications ranging from thermal energy storage to gas separation and water purification, but predominantly in heterogeneous catalysis[1,2]. The presence of aluminum, and in particular the necessary charge compensation add another layer of complexity to the structural characterization of these materials but are crucial to the catalytic function of zeolites. For example, the proton-exchanged aluminosilicate zeolites, i.e., Brønsted acidic site (BAS) zeolites, are one of the cornerstones of industrial petrochemical processes[3]. Recently, great experimental and theoretical efforts have been made to go beyond the traditional applications of zeolites, for example in converting sustainable bio-feedstocks into chemicals[4–6].

A further critical consideration for both existing and emerging applications is the interaction between BAS zeolites and water. This relationship governs many features of BAS zeolites including (i) proton solvation, and thus acidity[7,8], (ii) hydrolytic bond dissociation and defect formation, which controls catalyst durability and activity[9], (iii) water mobility and clustering in zeolite pores[10], and (iv) the synthesis of zeolites from precursor gels containing silica fragments, water and cations[11]. Owing to the microporous nature of zeolites, this interaction is not adequately viewed as a simple bulk-liquid interface, but rather a collection of complex binding, clustering, exchange, and reaction steps between variously sized water clusters and an inhomogeneous surface that is complicated by topology-dependent confinement effects[12]. As a result, the proper mechanistic understanding of BAS-water-zeolite interactions is still lacking, limited to either static calculations at ultra-high vacuum conditions[13,14] or exploratory (ab initio)

[1]Department of Physical and Macromolecular Chemistry, Faculty of Sciences, Charles University, Hlavova 8, 128 43 Prague 2, Czech Republic. [2]Mathematical Institute, Faculty of Mathematics and Physics, Charles University, Sokolovská 83, 186 75 Prague, Czech Republic. [3]Deceased: Petr Nachtigall. ✉e-mail: andreas.erlebach@natur.cuni.cz; lukas.grajciar@natur.cuni.cz

dynamical simulations of narrow scope[15–18]. These investigations demonstrate the importance of capturing dynamics under operating conditions, being able to discover unexpected reaction mechanisms and defective species that hitherto eluded structural identification, but are not sufficiently economical for a global exploration of structural and reactive space.

A standard tool for accelerating the reactive sampling in zeolite-water systems, and thus reaching experimentally relevant timescale or realistic levels of model complexity is the class of reactive analytical potentials, for example, ReaxFF[19]. However, due to their fixed functional form, these potentials have limited transferability to systems with different chemical composition[20]. Therefore, they frequently require re-parameterization for a specific system for fine-tuning[21]. An emerging alternative to analytical force fields is represented by machine learning potentials (MLPs), which interpolate the potential energy surface (PES) at the level of an ab initio training set[22–24].

Two paradigms dominate the MLP field currently: (i) training of universal MLP that cover large parts of the chemical space with dozens of elements, but with limited coverage of the configuration space, e.g., not considering all relevant chemical reactions with the associated transition states, e.g., OC22, CHGNet, and others[25–29], and (ii) active-learning procedures to develop system-specific MLPs to accelerate simulations for a specific model or thermodynamic state point[30–32], which capture the details of the PES, including transition states, but have little or no transferability to systems with different chemical composition. The universal MLPs, trained on very large datasets (few million data points), are constructed as highly transferable, but approximate models, meant for initial screening, typically with a need for further fine-tuning before being used for production runs. Also, commonly, the universal MLPs are trained on datasets comprising only close-to-equilibrium structures[25,27–29], hence caution is needed when applying them to activated events. On the other hand, the system-specific MLPs are able to achieve quantitative accuracy with respect to their reference level, even considering highly activated events, and being typically trained on only hundreds to low thousands of data points. However, their applicability is limited only to the specific system of interest. Hence, MLPs that are able to simultaneously cover the broad chemical and configurational space needed for a class of materials such as BAS zeolites, including the complexity of framework and water-framework-based highly activated reactive transitions, are currently missing.

In this work, we developed reactive global neural network potentials (NNP) for an entire material class, namely, BAS zeolites. The breadth of chemical and configurational space spanned by the training

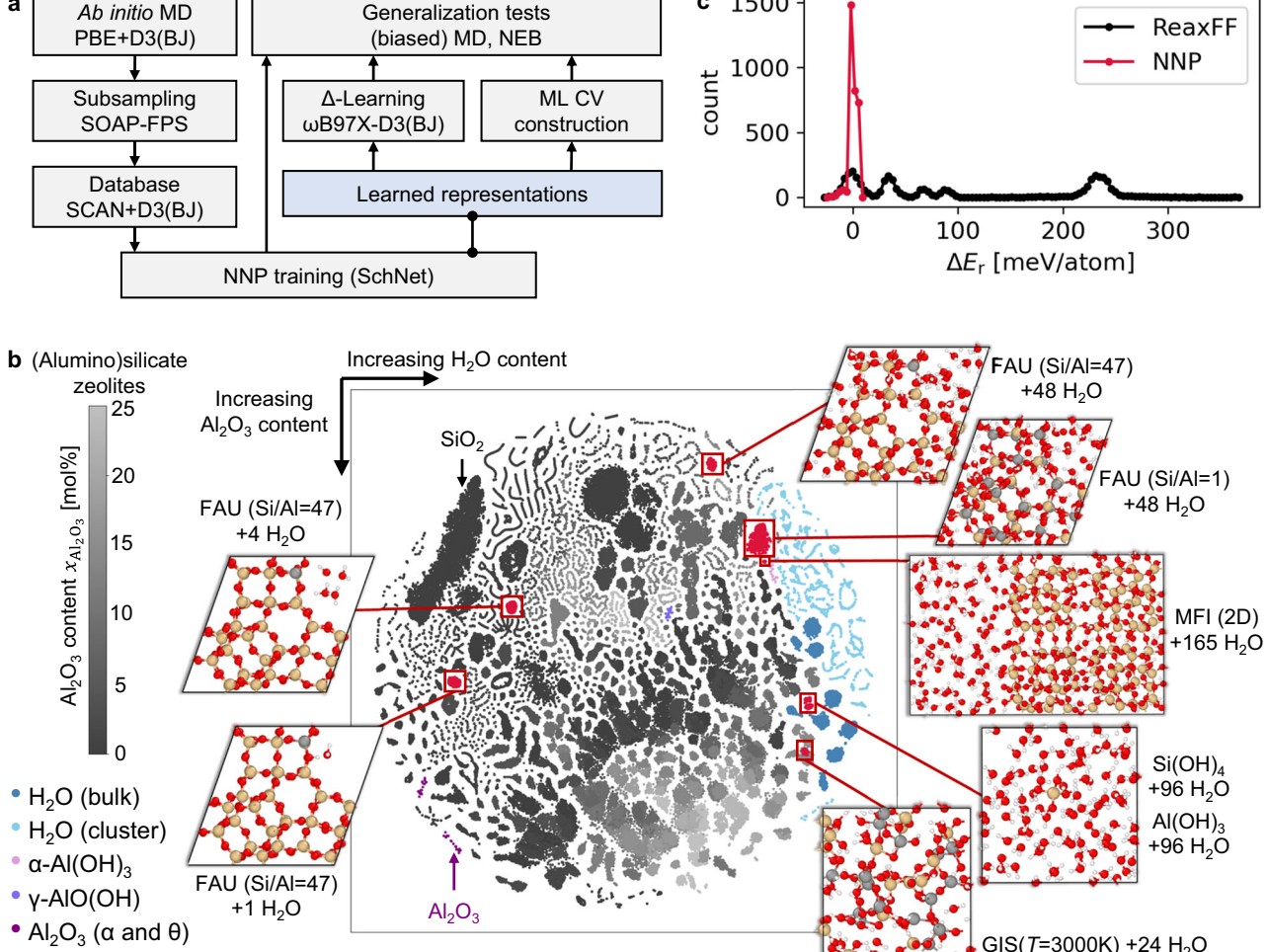

**Fig. 1 | Training and testing of general BAS zeolite NNPs. a** Computational workflow for the creation of the SCAN+D3(BJ) database using subsets of ab initio molecular dynamics (MD) trajectories selected by Farthest Point Sampling (FPS) with the smooth overlap of atomic positions (SOAP) descriptor. The generality of the NNPs was tested by (biased) MD and nudged-elastic band (NEB) calculations. The end-to-end learned representations are used for Δ-learning and construction of ML collective variables (ML CV). **b** t-distributed stochastic neighbor embedding (t-SNE) plot of the average representation vectors of all configurations in the training database (color codes shown on the left). Generalization tests are highlighted in red (Si: yellow, Al: gray, O: red, H: white). **c** Reaction energy error distribution $\Delta E_r$ (see Eq 1) of the NNPs in comparison with ReaxFF[58].

database (see Fig. 1 and Supplementary Figs. 1–2) as well as a consistently good performance of the NNPs in a battery of generalization and transferability tests considering multiple unseen zeolitic frameworks, various water loadings, and Si/Al ratios, indicates that these potentials are indeed able to capture large portions of the configurational and chemical space ranging from dense silica and alumina polymorphs, through water-containing BAS zeolites of varying Si/Al ratios, to bulk water and water gas-phase clusters. In addition, generalization tests showed hitherto unseen chemical species and processes, including a collective hydrolysis mechanism at the surface of a zeolite nanosheet. Finally, we show that the learned representations of the NNP baseline models can be used for data-efficient learning of higher (hybrid) DFT level corrections (Δ-learning)[33] for specific use cases, in addition to developing machine-learned collective variables for the acceleration of rare event sampling[34].

## Results

### Database generation and training of the general BAS zeolite NNPs

One of the challenges in training general NNPs for a material class is creating a training database that captures relevant parts of the configuration and chemical space. The computational procedure employed in this work is summarized in Fig. 1a. The bulk of the database is derived from 500 short (10 ps) ab initio molecular dynamics (AIMD) trajectories (at PBE + D3(BJ) level)[35–37] using a set of BAS zeolite models. This structure set contains 150 zeolites constructed using ten topologies (three existing and seven hypothetical) with varying Si/Al ratios (~1–32, only with Löwenstein pairs) and water loadings (from 0 to ~1.1 g cm$^{-3}$) at three temperatures ranging from 1200 K to 3600 K (see Supplementary Table 1). The seven hypothetical zeolite frameworks were selected from a siliceous zeolite database[38] by Farthest Point Sampling (FPS)[39,40] using the smooth overlap of atomic positions (SOAP)[41] kernel as metric to obtain configurations with structurally distinct atomic environments (see "Methods" section and Supplementary methods). We also added three existing zeolites with different framework densities (14.9–20.5 Si nm$^{-3}$) and a low-density, two-dimensional silica bilayer to further diversify the initial configurations in terms of atomic density and structure (see Supplementary methods). The chosen structures cover a broad range of zeolite ring topologies from 3-membered rings all the way 14-membered rings and are available under: https://doi.org/10.5281/zenodo.10361794. The AIMD simulations were performed on a wide temperature range (see above) to sample low- and high-energy parts of the PES which is a standard procedure to sample reactive, rare events in an unbiased way (see "Methods" section and Supplementary Methods)[38,42–44]. During these high-temperature AIMD simulations a number of distinct bond-breaking events were observed, such as Si–O/Al–O bond hydrolysis, generation of hydroxonium/hydroxide species or generation of other defective species with increased Al/Si coordination.

In addition, nine AIMD runs were performed for bulk water at three densities (0.9–1.1 g cm$^{-3}$) and at three temperatures (300–900 K). All AIMD trajectories were then subsampled by SOAP-FPS. We also subsampled (using SOAP-FPS) AIMD trajectories of zeolite CHA taken from our previous works on non-Löwenstein pairs (Al-O-Al) with various water loadings (0, 1, 15 water molecules)[45] and biased AIMD runs of Si-O(H) and Al-O(H) bond cleavage mechanisms[15]. These structures were used for SCAN+D3(BJ)[46] single-point (SP) calculations to create the bulk of the training database. We note that the enhanced sampling AIMD runs make up only a small portion of our dataset, as our aim was to allow for unbiased exploration of the phase space, allowing for the "discovery" of a priori unknown reactive mechanisms in such complex environments. However, the inclusion of the biased AIMD runs in the dataset is important and computationally efficient when generating training data for system-specific/reaction-specific MLPs[47–49]. To systematically sample states close to the equilibrium

structures, lattice deformations (see Supplementary Methods and Methods section) were applied to the optimized structures of the BAS zeolite models mentioned above, and these structures were then used for SCAN+D3(BJ) single-point (SP) calculations. Finally, to further diversify the database, the same lattice deformations were used for alumina and ice polymorphs as well as water clusters[50] (see "Methods" section).

The resulting database contains 248,439 structures which is much larger compared to datasets used typically to train MLPs for specific systems/reactions (e.g., those with fixed chemical composition)[30,31] yet much smaller than databases (e.g., OC22)[26] that aim to span the whole periodic table containing millions of DFT data points. We trained (an ensemble of six) SchNet-based potentials using a well-tested setup of hyperparameters[38,48,51] which provides the best performance also herein (see Supplementary Fig. 3 and "Methods" section for details on hyperparameter testing). The trained NNPs achieve an average test root mean square errors (RMSE) of 5.3 meV atom$^{-1}$ and 186 meV Å$^{-1}$ for energies and forces, respectively (see Supplementary Table 2). These errors are similar to other reactive (rotationally invariant) MLPs[42,44,52] and about the same as for our previously developed silica NNPs[38]. Note also that such errors are not larger than the errors between two different "flavors" of the (meta)GGA exchange-correlation DFT functionals (see, e.g., Supplementary Table 4 in ref. 38 comparing the energy and force errors of PBE and SCAN functionals). This indicates that the trained NNPs are expected to retain the (meta)GGA DFT level of accuracy. With the adoption of the new generation of rotationally equivariant NNPs one can expect the training errors to drop few-fold[53–55], getting the trained NNPs closer to a specific reference level used for their training, i.e., the SCAN-D3(BJ) level herein. Nevertheless, the performance tests of the herein-trained NNPs for reactive events and various near-equilibrium properties in this work (see below) show that these errors do not compromise the value of the potentials for application to zeolite problems, retaining the intended (meta)GGA DFT quality.

Figure 1 b provides a low-dimensional representation of the training database. It shows a t-distributed stochastic neighbor embedding (t-SNE)[56] plot of the averaged SchNet representation vectors (see Supplementary methods) to visualize the structural and chemical diversity of the training database, ranging from water-free alumina and silica systems through various water-loaded BAS zeolites to bulk water and small clusters. The t-SNE components of the averaged SchNet representations change smoothly with the chemical composition of the structures as well as with their total energy (see Supplementary Fig. 2). In addition, all generalization tests (see below) lie within the generated interpolation grid of the database and cover a wide range of application cases for zeolite modeling.

### Generalization tests and exploration of configuration space

To properly test the generalization abilities of the trained NNPs, we employed a series of simulations for systems outside of the training domain (OOD or generalization tests), i.e., including: (i) MD simulations at ambient conditions that probe the performance of NNPs for close-to-equilibrium structures and (ii) high-temperature MD simulations (supplemented by nudged-elastic band (NEB) transition path searches) to assess the NNP quality for highly activated, reactive events (see "Methods" section). In these MD simulations, the NNP forces were used to propagate the systems in time and DFT single-point (SP) calculations were carried out for configurations uniformly sampled from the MD trajectory. In addition, we have carried out further validations (most of them of OOD nature), in which we have driven the calculations (biased and equilibrium MD as well as geometry optimization) using both the DFT and NNP forces and compared their performance on an equal footing for multiple properties (densities, lattice parameters, adsorption energies, free energy profiles, etc.) - we defer further discussion of these validations to the sections "Sampling

equilibrium properties" and "NNP robustness at high temperatures" below. The systems considered in generalization tests sample the chemical and structural space of (water-loaded) BAS zeolites (see Fig. 1b) varying in water and aluminum content, as well as in the zeolite topology (FAU, GIS, and MFI zeolite frameworks which are not seen during the training) and dimensionality of the BAS systems (three-dimensional crystal, zeolite layer or a zeolitic molecular fragment interacting with bulk water). Here, we focus on the overall performance of our trained NNP in these generalization tests. To quantify the NNP accuracy and generalization capability, we use two energy error metrics to compare NNP (and ReaxFF) energies with DFT: a reaction energy error $\Delta E_r$ and a relative energy error $\Delta\Delta E$. Firstly, we define the reaction energy $E_r$ of the hypothetical formation reaction:

$$x\text{SiO}_2 + \frac{y}{2}\text{Al}_2\text{O}_3 + \frac{z}{2}\text{H}_2\text{O}_{(g)} \rightarrow \text{Si}_x\text{Al}_y\text{H}_z\text{O}_{2x+1.5y+0.5z} \quad (1)$$

with $\alpha$-quartz, corundum ($\alpha$-Al$_2$O$_3$), and a single water molecule in the gas phase as reference structures. This reaction energy is used for calculation of the NNP (and ReaxFF) error $\Delta E_r$ with respect to the DFT reference level (here: SCAN+D3(BJ)) using the DFT SP calculations of the subsampled NNP trajectories. Adoption of $\Delta E_r$ allows for benchmarking methods across a broader (BAS zeolite) chemical space, as exemplified by Hautier et al.[57]. Alternatively, we also used relative energies $\Delta E$ with respect to a reference configuration with the same chemical composition, e.g., the initial structure of an MD trajectory or NEB calculation (see Supplementary Table 3), to define the relative energy error $\Delta\Delta E$. This metric only quantifies energy errors within the same system but not across the chemical space. Obviously, the force errors as intensive properties are independent of the reference and can be compared directly across the chemical space. Table 1 summarizes the RMSEs of energies and forces for all test cases (2700 structures in total) of the NNP with respect to SCAN+D3(BJ) reference and Fig. 1c shows the total energy error $\Delta E_r$ distribution (see Supplementary Fig. 4 for $\Delta E_r$ and $\Delta\Delta E$ distributions for each system separately). Fig. 1c (and Supplementary Table 3) also show the performance of the standard reactive analytical force field ReaxFF specialized for water-loaded BAS zeolite systems[58].

The total NNP errors are similar to other state-of-the-art (rotationally invariant) MLPs[38,42,44] for the modeling of reactive events. More importantly, the NNP calculated reaction energies $E_r$ are consistent over the entire range of chemical BAS zeolite compositions and configurations. Only in the case of GIS($T = 3000$K) + 24H$_2$O, the energy and force errors about twice as high compared to the other test cases. Such higher errors were also obtained for MLPs when applied to simulations with a large number of reactive events at extreme temperatures[38,42]. To put these NNP errors in context, note that standard GGA-level DFT functionals show, for 27 formation reactions that involve silica, an RMSE of about 28 meV atom$^{-1}$ with respect to experiment[57]. Therefore, the NNPs safely retain the metaGGA-level DFT

quality for the description of the water-loaded BAS zeolite systems. In addition, we have also tested, whether the NNP reproduces the DFT potential energy surface consistently across the configurational space. For this purpose, we evaluated the covariance between the force (energy) errors and the total forces (and energies) from the explicit DFT calculations (see Supplementary Table 3). The average covariance (averaged overall generalizations tests) between the actual forces (at DFT) and the NNP error in forces is small (0.13), which indicates that errors in forces are mostly random (the largest force covariance was obtained for FAU(Si/Al = 1) + 48H2O system reaching 0.24), and largely independent of the configurations sampled. This shows that the NNP is not expected to bias the sampling of the configurational space, representing correctly the (average) curvature (structure) of the reference potential energy surface. We also note that MLPs with about the same force accuracy were shown to accurately reproduce equilibrium properties such as vibrational density of states[38,42].

**Sampling of equilibrium properties.** The dynamic behavior of zeolite-confined water-containing solvated protons is of high interest due to the (potential) applications of zeolites in water purification[10,59], heat storage[60] or reaction optimization under humid conditions (such as biomass conversion)[7]. The ability to realistically model the (water-loaded) BAS systems close-to equilibrium is crucial for understanding of many of their signature properties such as acidity, (water) adsorption and diffusion, or relative stability as a function of topology, water content, and aluminum distribution and concentration.

The role of water loading and aluminum concentration was probed using equilibrium MD simulations of water-loaded zeolite with FAU (faujasite) topology (see "Methods" section for details about the model generation) - an industrially important zeolite topology unseen in the NNP training - under standard conditions (300 K). We considered model systems with the (theoretically) lowest and highest possible Si/Al ratios in a (primitive) FAU unit cell, namely, a single Brønsted acid site (BAS) with Si/Al = 47 and Si/Al = 1 according to Löwenstein's rule that prohibits the formation of Al–O–Al pairs. In the case of Si/Al = 47 (FAU(Si/Al = 47) + $n$H$_2$O), three water loadings $n$ were tested, from single water through a water tetramer to full water loading of FAU with 48 water molecules (approximate water density of 1 g cm$^{-3}$). For Si/Al = 1, full water loading with 48 molecules per FAU unit cell was chosen to focus on extensive sampling of BAS protonation and deprotonation events, a key reactive event characterizing these strong solid acids.

In the case of the FAU (Si/Al = 47) model, the single water molecule ($n = 1$) remains adsorbed at the BAS throughout the 1 ns MD simulation, in line with the very strong interaction between BAS and water molecule. This strong binding is characterized by the water adsorption energy of approx. −79 kJ mol$^{-1}$ calculated here using NNP (evaluated as an average over the MD trajectory) - this strong stabilization is in qualitative agreement with the static single water adsorption energies in CHA zeolite reported in the literature ranging from −50 to almost −100 kJ mol$^{-1}$ depending on the type of exchange-correlation DFT functional adopted[61]. We also quantified the degree of solvation by calculating the minimum distance of Al-O$_{FW}$-Si framework oxygens to all hydrogen atoms (see Supplementary Fig. 5). The proton was considered solvated if it is closer to a water oxygen than to Al-O$_{FW}$-Si. Only very few solvated states (less than 3% of the trajectory) were observed during the 1 ns run, in line with previous (shorter) AIMD simulations at the DFT level for FAU and CHA zeolite[17,45]. However, the water tetramer ($n=4$) is already able to deprotonate the BAS, but similarly to single water, the tetramer stays close to the framework Al during the 1 ns MD trajectory (on average 3.1 Å between Al and the cluster center-of-mass, see Supplementary Fig. 5). At full water loading ($n = 48$), the proton rapidly leaves the BAS and stays solvated with an average distance of 7.3 Å (ranging from 3 to 10 Å) from the Al-O$_{FW}$-Si (see Supplementary Fig. 5). Importantly, both the degree of

**Table 1 | Root mean square errors of the NNP reaction energies $\Delta E_r$ (see Eq 1) [meV atom$^{-1}$] and forces [eV Å$^{-1}$] with respect to the SCAN+D3(BJ) reference for the test cases shown in Fig. 2**

| Generalization test case | Energy | Forces |
|---|---|---|
| FAU(Si/Al = 1) + 48H$_2$O | 0.9 | 0.11 |
| FAU(Si/Al = 47) + $n$H$_2$O | 2.4 | 0.07 |
| Si(OH)$_4$ + 96H$_2$O | 4.1 | 0.12 |
| Al(OH)$_3$ + 96H$_2$O | 4.1 | 0.12 |
| GIS($T = 3000$K) + 24H$_2$O | 10.0 | 0.28 |
| MFI(2D) + 165H$_2$O | 1.4 | 0.16 |
| Average | 3.8 | 0.11 |

deprotonation and the distance of the proton from the BAS as a function of water loading are in very good agreement with the biased AIMD simulations of Grifoni et al. for protonic FAU zeolite[17]. However, the evaluation of the confined water dynamics herein is hindered by finite-size effects, due to a small primitive cell of FAU chosen primarily for benchmarking purposes. Hence, we refer the interested reader to our preliminary work[62] using the NNPs presented here on water diffusion in FAU using an appropriately sized FAU unit cell (cubic cell with edge length of 25 Å) that is prohibitively large for carrying out routine DFT calculations. A particularly challenging case is the interaction of water with the framework of FAU with Si/Al = 1, owing to the large number of BAS and complex water-mediated communication between acid sites. Despite the fact that the MD trajectory contains several protonation and deprotonation events, the errors of the NNP for this challenging case increase only mildly (see Table 1) and are well below the test errors for the training database (see above). Thus we conclude that the NNP is able to reproduce the solvation behavior of zeolites across a wide space of Si/Al ratios and water loadings.

To check the NNP generality further, we employed the "inverse" models to water-loaded zeolites, namely the fragments of zeolites (silicic acid $Si(OH)_4$ and aluminum hydroxide $Al(OH)_3$) solvated in bulk-like water (using a simulation box containing 96 water molecules). Such systems are relevant for modeling potential precursors of zeolite synthesis or products of zeolite degradation in hot liquid water (de-silication and de-alumination)[5,9,63,64]. Since both processes take place under rather harsh hydrothermal conditions (temperature above 100 °C and pressure above 10 bars), the test simulations were carried out at 500 K. In both cases, the NNPs accurately reproduce the SCAN +D3(BJ) energies and forces. Similar to the previously discussed system $FAU(Si/Al = 1) + 48H_2O$, the NNP energy errors (4 meV atom$^{-1}$) are mainly connected to the offset of $E_r$ (see Supplementary Fig. 4).

Besides the generalization tests above, which were based on NNP-driven MD simulations subsampled with DFT single points, we also considered three other OOD tests, which compared DFT-driven simulations (MD and geometry optimizations) with the NNP single points and geometry optimizations. First, following the geometry optimizations both at the reference DFT (SCAN+D3(BJ)) and the NNP level we evaluated their performance in a "head-to-head" fashion for multiple properties (energies, densities, average Si-O bond distances, lattice parameters) of the small set of purely siliceous zeolites for which the corresponding experimental data are known (see Supplementary Table 4). The average results (based on mean absolute deviation (MAD)) show that the NNP provides essentially identical values to the reference DFT for lattice parameters and average Si-O bond distances, slightly underestimates the density (by 0.1 Si nm$^{-3}$ mostly due to a poorer description of the dense silica polymorphs) and slightly improves (compared to experimental data) the relative stabilities (by 0.7 kJ mol$^{-3}$ Si). This dataset allowed us also to compare the performance of the current NNP to our original NNP trained only using all-silica data[38]; despite covering broader chemical space, the current NNP show similar performance, which we relate to a more diverse set of silicon and oxygen local atomic environments (including those of all-silica nature) included in the current database. Next, we tested how sensitive is the NNP to a mildly different structural nature of symmetrically nonequivalent aluminum sites in an OOD MFI framework in protonic form (Si/Al = 95), which we probed by evaluating single water adsorption energies on all twelve (T1-T12) symmetrically nonequivalent aluminum sites (see Supplementary Fig. 7). The NNP water adsorption energies range between −80 and −110 kJ/mol, are characterized by MAD of about 10 kJ mol$^{-1}$ with respect to the SCAN+D3(BJ) reference and the NNP in most cases follows the relative ordering of stabilities observed at the reference level. To put this performance into context, the MAD of another DFT exchange-correlation (XC) functional (PBE+D3(BJ)) with respect to SCAN+D3(BJ) is 12 kJ mol$^{-1}$, i.e., the NNP is at least as good as an approximation to the SCAN+D3(BJ) as

another "flavor" of the XC DFT functional. Lastly, we carried out MD simulations at the reference DFT level for "pinned hydroxonium" species (see Supplementary Fig. 6). These species were observed to form during the NNP MD simulations of $FAU(Si/Al = 1) + 48H_2O$ and in similar NNP-based simulations before[62]. The "pinned hydroxonium" species remained stable during 10 ps long SCAN+D3(BJ) AIMD, the NNP errors both in energies and forces remain rather small (similar to the overall RMSE for the whole database). Besides validating the NNP, these results also demonstrate the capacity of the NNP to explore hitherto unseen species, such as the "pinned hydroxonium", which may indeed be present in zeolites with very high Al content. In conclusion, the above-presented tests strongly suggest that the NNPs developed herein enable us to run reliable large-scale equilibrium MD simulations across broad parts of the chemical and configuration space of BAS zeolites and water with a level of accuracy that is close-to metaGGA DFT.

**NNP performance for highly activated reactive events.** Modeling of chemical reactions at the BAS zeolite-water interface requires a robust interpolation of the relevant transition states. However, MLPs are expected to have only limited capability to reliably describe configurations and energetics in extrapolated or sparsely interpolated regions of the potential energy surface[22], which often coincide with the high-energy transition state configurations. Therefore, we tested the trained NNPs by performing MD simulations at very high temperatures (at 1600 and 3000 K) for an unbiased assessment of the NNP quality and robustness for modeling reactive processes. We note, that the very high-temperature conditions are not applicable to standard applications of water-zeolite systems, however, the reactive (rare) events observed with increased probability at such conditions are expected to be both realistic (see below) and a challenging case to model accurately. We chose two systems that were not part of the training dataset: an interface model comprised of a siliceous MFI slab in interaction with bulk-like water and water-loaded GIS with Si/Al = 1. In addition, we tested the accuracy of the NNPs using static nudged-elastic band (NEB) calculations, which are used to locate specific transition pathways, for four reactions relevant for BAS zeolite-based catalysts (see below). Lastly, we evaluated the free energy barrier for a proton jump in the water-free CHA model (an in-domain case) both at the NNP and the reference SCAN+D3(BJ) level.

The first generalization test was a model for the external interface of siliceous MFI with bulk water (see "Methods" for details about the slab model employed). This zeolite model resembles 2D-MFI nanosheets which have been successfully prepared by exfoliation of a multi-lamellar MFI zeolite[65]. To sample reactive events at the external zeolite surface, we performed an exploratory MD run at 1600 K for 1 ns. No extrapolation was detected using the ensemble of NNPs. As expected, significantly increased temperature leads to increased probability of the highly activated reactive events to take place and we do observe a relevant chemical reaction taking place over the course of 5 ps, in which a silanol defect is created at the external MFI surface (see Fig. 2a). In our previous work[15], a fairly similar Si-O bond-breaking mechanism involving proton transfer was found to be characterized by a free energy of activation amounting to approx. 80 kJ mol$^{-1}$ (at 450 K). The observed reaction starts at the intersection of the bulk water phase with the MFI main channel (along the crystallographic $b$-direction). Firstly, a water molecule adsorbs at a surface Si site (Fig. 2b), which is a standard precursor for Si-O bond hydrolysis reported in multiple publications before[15,66–68]. The autoprotolysis of a nearby water molecule leads to the transfer of a proton from the adsorbed water molecule to the formed hydroxide ion creating an additional surface silanol group at the five-fold coordinated Si atom (Fig. 2c). The remaining hydroxonium ion shuttles the excess proton together with the surrounding water molecules to a framework oxygen bound to the five-fold coordinated Si (Fig. 2d). This process finally leads to the cleavage

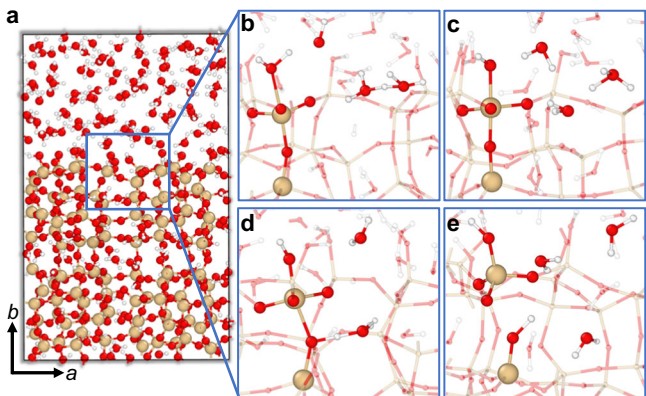

**Fig. 2 | Surface defect creation in an 2D-MFI nanosheet. a** Snapshot of 2D-MFI from an exploratory 1 ns MD run at 1600 K to sample reactive events (Si: yellow, O: red, H: white). **b–e** Reaction steps of silanol defect creation at the 2D-MFI-water interface: **b** water adsorption on a surface Si and water autoprotolysis; **c** proton transfer from the adsorbed water to the hydroxide ion; **d** migration of the hydroxonium ion and adsorption on a framework oxygen; **e** Si–O bond hydrolysis, creating a silanol defect in axial position to the formed surface silanol.

of the Si−O(H) bond, creating a silanol defect in an axial position to the previously formed surface silanol, i.e., the same product and mechanism observed previously to take place inside the cage systems of various zeolites (CHA[15], UTL[18]). Our exploratory MD simulation revealed a feasible reaction mechanism for silanol defect creation at the external zeolite surface involving the autoprotolysis of water, which would be challenging to find with biased dynamics simulations with human-designed CVs. These findings are also in line with previous experimental studies on the hydrolysis of MFI zeolites in hot liquid water that suggest that zeolite degradation predominantly starts at the external surface, in which water autoprotolysis and silanol groups play a crucial role[69,70]. To confirm that the defect creation process observed at the NNP level is reliable, we performed SCAN+D3(BJ) SP calculations using 100 snapshots comprising the reaction steps depicted in Fig. 2b−e. The NNPs proved very accurate for this test case with an energy RMSE of 1.4 meV atom$^{-1}$ (see Table 1).

The second particularly challenging generalization test was a GIS zeolite model (Si/Al = 1) loaded with waters (24 molecules), which was molten at 3000 K for 2 ns to sample multiple highly activated reactive events taking place simultaneously (see also Supplementary Fig. 8). We obtained a stable MD trajectory of the liquid BAS zeolite state with thousands of bond-breaking events (including Si−O and Al−O bond hydrolysis, aluminol, and silanol formation, increase of Al coordination up to six, water splitting/re-formation, water autoprotolysis, (de)protonation of the silanols/aluminols and formation of aluminum oxide-like islands in the framework) over the entire simulation time, without detecting extrapolation using the trained ensemble of NNPs. However, this test case shows higher energy RMSE by around a factor of two when compared to the other test cases (see Table 1). Similar trends of increased energy errors have also been observed for other MLPs applied to high-temperature MD runs of the liquid state of strongly (covalently) bound materials (see e.g., Refs. 38,42). Even though the NNP accuracy mildly deteriorates at these extremely high temperatures, they proved robust in these simulations of a large variety of highly activated reactive events.

Next, we tested the accuracy of trained NNPs on specific elemental reactions in water-loaded BAS zeolites with well-known transition states: a proton jump with and without water[48,71,72], and water-assisted bond-breaking mechanisms of the Si−O and Al−O(H) bonds[14,66,73] (see Supplementary Fig. 9 and Supplementary Table 5 as well as Fig. 3). All NEB calculations were performed for FAU; an

industrially relevant zeolite topology that was not part of the training dataset. For quantification of the NNP error, we used SCAN+D3(BJ) SP calculations on all generated NEB images. On average, the relative NNP energies only slightly deviate from their DFT reference with an RMSE of about 6 kJ mol$^{-1}$. Such small errors for activation barriers can be considered to lie within DFT accuracy which is on average about 20−30 kJ mol$^{-1}$ with respect to coupled cluster calculations in the case of (meta)GGA functionals[74].

Lastly, we calculated, using metadynamics simulations[75], the free energy profile of a proton jump process (between O2 and O3 oxygens) in the water-free CHA model (see Fig. 3a). The NNP is able to reproduce the reference SCAN-D3(BJ) DFT free energy profile very well, with only minor deviations (within 10 kJ/mol for both free energy barriers and reaction energies). Note, that such minor deviations are expected to lie well within the errors that are due to the sub-optimal set-up of the free energy method parameters (see Supplementary Fig. 12 or a related recent work[76]). These tests strongly suggest that the herein-developed NNP indeed reconstructs broad parts of the DFT-based PES well and is thus able to sample similar configurational/phase space as the DFT reference.

We also note that many of the reactions probed above involve hydrogen. Hence the nuclear quantum effects (NQEs) will affect the dynamics and barriers. A recent work from Bocus et al.[48] attempted to quantify these effects for proton jump in H-CHA zeolite reporting the lowering of the proton jump barriers by a rather small amount (5–10 kJ/mol) depending on the temperature and we expect similar minor effects in our simulations. However, quantifying the extent of NQEs is out of the scope of the current contribution.

In conclusion, the tests presented in this section indicate that the trained NNPs are robust and general interpolators for simulations across broad parts of the chemical and configuration space spanned by the water-loaded BAS zeolites and that they are able to retain the reference-level (SCAN+D3(BJ)) DFT quality not only for close-to-equilibrium simulations but also for highly activated reactive events.

## Extensions of the NNP model

Obtaining a general NNP model that describes water-loaded BAS zeolitic systems with metaGGA DFT quality with several orders of magnitude speedup is clearly beneficial. However, with such a robust baseline NNP model available, it is possible to construct extensions that can improve either the accuracy of the description or the efficiency of sampling of the reactive events of interest.

**Improving the baseline model accuracy using Δ-learning.** To improve the accuracy of the benchmark level, one can employ the well-known Δ-learning concept[33]. In this way, one may train a correction model on top of the baseline model, using a computationally more demanding but more accurate level of theory for a small subset of data points that typically covers only the specific process/reaction of interest (e.g., a proton jump in a specific zeolite framework)[48]. Δ-learning is typically computationally much more efficient than training of an NNP directly using only data from a high-level method. The reason is that the correction- (Δ-) surface is expected to be much smoother than the full potential energy surface. For the higher-level reference, we chose the range-separated hybrid DFT functional ωB97X complemented with the empirical dispersion correction D3(BJ): a functional that shows considerably better performance for water cluster binding energies and reaction barriers[74,77,78] than our baseline reference SCAN+D3(BJ) functional.

First, we considered a common application domain/target of the Δ-learning approach, in which one trains and deploys the correction (ΔNNP) model on the same reaction/process (the in-domain case)[79], i.e., the proton jump in CHA. Initially, we generated a small ωB97X-D3(BJ) database containing 500 structures taken from the biased (NNP level) MD runs of an H-jump in water-free CHA (between O2 and O3,

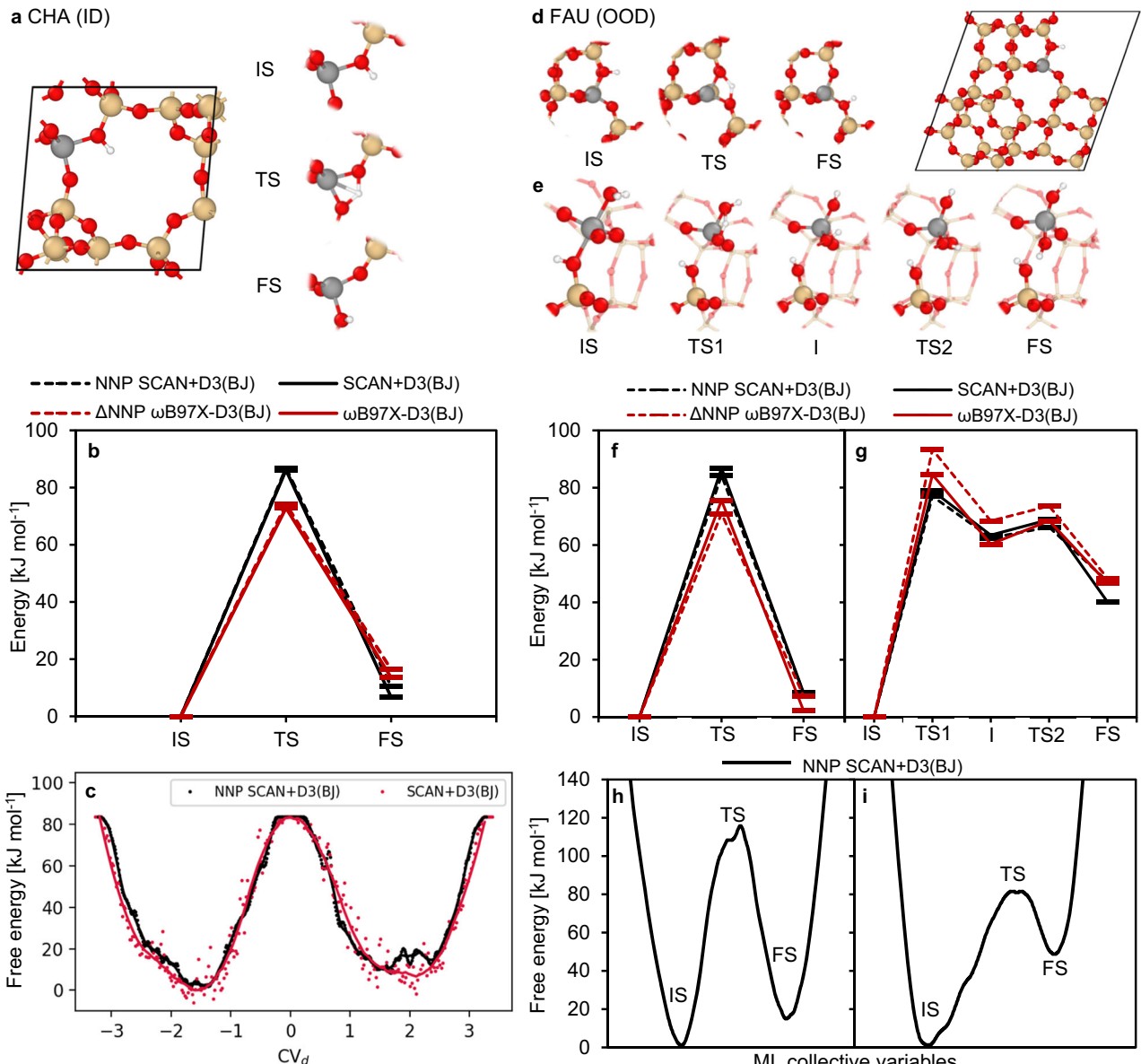

**Fig. 3 | Reaction path modeling using Δ-learning and ML collective variables.** Reaction path modeling for an (Δ)NNP in-domain (ID) case: a proton jump in CHA (**a**–**c**); and two out-of-domain (OOD) cases: a proton jump (**d**, **f**, **h**) and an Al–O(H) bond dissociation (**e**, **g**, **i**) in FAU. **a**, **d**, **e** Atomic structures along the reaction path (Si: yellow, Al: gray, O: red, H: white). **b**, **f**, **g** Static (Δ)NNP simulations and corresponding DFT energies for the NNP baseline level (SCAN+D3(BJ)) and the ΔNNP level (ωB97X-D3(BJ)). **c**, **e**, **f** Estimated free energy profiles using a standard collective variable $CV_d$ (see "Methods" section) for CHA and using ML collective variables for FAU.

see "Methods" section) taken from Ref. 34. Next, we trained a correction (ΔNNP) to the atomic energies of the NNP baseline model by using a simple linear regression for the ΔNNP model (see Method section). It was found that 150 training structures were sufficient to reach (test set) RMSEs of 1.3 meV atom$^{-1}$ and 69 meV Å$^{-1}$ for energy and forces, respectively (see Supplementary Fig. 10). Figure 3b show the results of the NNP level NEB calculations along with the corresponding DFT energies. Figure 3a depicts the structures for the reaction path and Table 2 also compares the relative energies of the proton jump in CHA (O2–O3). The deviation between the reference ωB97X-D3(BJ) and the (baseline plus) ΔNNP model is very small (less than 3 kJ/mol), i.e., within chemical accuracy (less than 1 kcal/mol).

Next, we focused on more challenging cases, in which we tested the performance of the ΔNNP model in two out-of-domain (OOD) scenarios - first, the same type of reaction (proton jump) in an OOD framework (FAU) and second, an OOD reaction in an OOD framework

(Al-O(H) bond dissociation in FAU). In particular, we chose FAU with a single Al site (Si/Al = 47) as a test case. Figure 3d–g and Tables 2–3 show the results of the NNP level NEB calculations for these OOD tests along with the corresponding DFT energies. For the proton jump in FAU the ΔNNP accuracy of the relative energies mildly deteriorates (about

**Table 2 | Relative energies Δ*E* [kJ mol⁻¹] of the proton jump in CHA and FAU at the (Δ)NNP and DFT level**

|  | CHA | | | | FAU | | | |
|---|---|---|---|---|---|---|---|---|
|  | SCAN +D3(BJ) | | ωB97X-D3(BJ) | | SCAN +D3(BJ) | | ωB97X-D3(BJ) | |
|  | NNP | DFT | ΔNNP | DFT | NNP | DFT | ΔNNP | DFT |
| TS | 87 | 86 | 74 | 73 | 84 | 87 | 71 | 76 |
| FS | 10 | 7 | 16 | 14 | 8 | 8 | 7 | 2 |

**Table 3 | Relative energies ΔE [kJ mol⁻¹] of the Al-O(H) bond dissociation in FAU at the (Δ)NNP and DFT level**

|       | SCAN+D3(BJ) | | ωB97X-D3(BJ) | |
|-------|-----|-----|------|-----|
|       | NNP | DFT | ΔNNP | DFT |
| TS1   | 77  | 79  | 93   | 85  |
| I     | 63  | 63  | 68   | 60  |
| TS2   | 66  | 69  | 74   | 68  |
| FS    | 47  | 40  | 48   | 47  |

5 kJ mol⁻¹ error) compared to the CHA case, however, it still provides an improved description compared to the baseline NNP model with the baseline reference-level metaGGA DFT. For the most challenging OOD case of the Al−O(H)−Si bond scission in FAU (see Method section for details), the error of the ΔNNP model with respect to the reference ωB97X-D3(BJ) in the relative energies ΔΔE further mildly increases (up to 8 kJ mol⁻¹) (see Table 3). Unfortunately, for this particular OOD case, such an error in relative positioning of the stationary states is already close to the differences between baseline- (SCAN+D3(BJ)) and the high-level (ωB97X-D3(BJ)) DFT predictions (within 7 kJ/mol), making the ΔNNP model predictions, in this particular case, as good (or as bad) as the those of the baseline NNP model. However, we note that the fact that the ΔNNP model, in conjunction with the baseline NNP does not worsen the description over the baseline NNP for such a challenging OOD case is a surprising result.

A more in-depth analysis of the ΔNNP errors for these OOD cases is provided in the Supplementary Information (see Supplementary Table 6), including an extended dataset containing also the single-point data taken from biased dynamics runs depicted in Fig. 3h, i (Supplementary Fig. 11 and Supplementary Table 7). In summary, ΔNNP simulations can model chemical reactions similar to those covered by the training data, even in an OOD zeolite framework, but has only limited generalization capability to systems with different chemical composition (see Supplementary discussion). Nevertheless, the fairly good extrapolation robustness of our (very simple) ΔNNP model for similar reaction pathways indicates both that the general NNP baseline model is rather robust across the BAS zeolite PES (with water) and that the correction surface is rather simple (low-dimensional) as, e.g., shown recently for the DFT-to-MP2 correction surface of alkane adsorption in protonic zeolites[80].

**Accelerating rare event sampling using baseline model representations.** In the previous section, we tested the Δ-learned model for accurate (hybrid DFT) modeling using static calculations of known reaction mechanisms. However, prior investigations have shown the unforeseen and highly collective nature of water-involved reaction mechanisms as well as the sizable role of temperature effects[15,16,18]. Both imply the need for a tool with the ability to effectively discover and sample transition pathways. For effective sampling of the activated reactive events, which are therefore rare on the timescales accessible even for the NNP-accelerated simulations, one typically adopts a biasing along a low-dimensional representation of the reactive process, i.e., along the reaction coordinates or collective variables (CVs). However, good CVs can be difficult to construct in case of unknown, possibly complex, reaction pathways.

Our recent work[34] shows how the end-to-end learned atomic representations of our baseline NNP model can be used to automatically generate robust machine-learned CVs (ML-CVs). In this approach, the structures of the reactant, product, and perhaps also tentative transition states are first represented using the atomic representations of the herein-trained baseline NNP model. Next, these representations are used as an input for the dimensionality reduction model (variational autoencoder), which generates a low-dimensional (typically one- or two-dimensional) latent space from which the model

attempts to reconstruct the input representation vectors as precisely as possible. As a result, the latent low-dimensional space effectively distinguishes products from reactants, i.e., it represents the reactive coordinate or collective variable.

We showed previously that learned ML-CVs coupled with the baseline NNP enable efficient sampling of the free energy surface for a proton jump and Si-O bond hydrolysis in CHA zeolite[34]. Here, we test this procedure using the aforementioned proton jump and Al-O2(H) bond dissociation mechanism in FAU with Si/Al = 47 which is outside of the NNP training domain, in contrast to CHA (see "Methods" section). Figure 3h, i show the estimated free energy profiles calculated with ML-CVs using well-tempered metadynamics[75] simulations (see Methods section). The free energy barrier (approx. 110 ± 10 kJ mol⁻¹ at 300 K) of the proton jump in FAU is somewhat higher compared to the static calculations (84 kJ mol⁻¹, see Table 2). This is in line with previous calculations[48,72] which showed increasing reaction barriers with temperature (up to 20 kJ mol⁻¹ from 0 K to room temperature). In case of the Al-O(H) bond dissociation the activation free energy is about 80 kJ mol⁻¹, similar to the barrier found by the NEB simulations (see Table 3). Hence, with the baseline NNP model, one can not only accelerate the evaluation of energies (and forces) necessary for sampling the water-loaded BAS zeolite systems but also use it to automatically generate ML-CVs accelerating the sampling of a particular reactive process.

In this work, we developed a neural network potential (NNP) to cover the entire class of proton-exchanged aluminosilicate (BAS) zeolites, which are one of the cornerstones of existing petrochemical processes[3], as well as one of the main candidates for emerging applications in sustainable chemistry[2]. The breadth of chemical and configurational space spanned by the training database as well as a consistently good performance of the NNPs in a battery of generalization and transferability tests suggests that our NNP is able to provide a general approximation of the potential energy surface of the BAS zeolites, including reactive interactions with water, capturing both close-to-equilibrium structures and high-energy bond-breaking scenarios. These tests ranged from zeolite surfaces varying in water and aluminum content to zeolite fragments solvated in bulk-like water and a high-temperature melt of the aluminosilicate zeolite GIS. The very good performance of the NNP points to the high transferability of the NNPs, which are able to maintain consistent accuracy close to the (meta)GGA DFT level, outperforming standard analytical reactive force fields for water-loaded BAS zeolites[58] by at least one order of magnitude. Moreover, in some of these tests, we observed hitherto unseen chemical processes and species, which confirms the capability of the NNP for exploration and discovery of reactive pathways, in addition to the acceleration of configuration space sampling.

Furthermore, we exemplified on a small set of use cases, how the herein-developed NNP can be used as a basis for further extensions/improvements such as: (i) data-efficient adoption of higher-level (range-separated hybrid DFT) description via Δ-learning[33], and (ii) acceleration of reactive event sampling using automatic construction of collective variables, via end-to-end learned atomic representations[34]. Hence, we have highlighted how the baseline NNP model with its ML-based extensions may constitute a broader ML-based framework within which one can simulate BAS zeolite materials in a comprehensive, bias-free fashion with tunable accuracy.

We expect that the NNP, especially when complemented with the extensions exemplified above will represent a big step towards large-scale simulations of BAS zeolites tackling long-lasting challenges in the field, ranging from understanding the mechanistic underpinnings of zeolite hydrothermal (in)stability to the determination of the character of active species and defects under operating conditions. The application of these methods to long-time-scale diffusive processes and reaction networks in highly defective model systems that match those

of industrial application is already underway, with more data-efficient rotationally equivariant NNP architectures being adopted.

## Methods

### Dataset generation

Covering the chemical and configuration space of BAS zeolites requires a structurally distinct set of zeolite frameworks with different water loadings and Si/Al ratios. In our previous publication[38], we used SOAP-FPS[39–41] to find a subset of siliceous zeolites that optimally covers the structural diversity of existing and more than 300 k hypothetical zeolites. From this subset, ten zeolites were selected, three existing (CHA, SOD, MVY) and seven hypothetical zeolite frameworks (see Supplementary methods). These frameworks were used for the construction of 150 initial structures combining four water loadings (from 0 to-1.1 g cm$^{-3}$) with three Si/Al ratios between -1–32 (in protonic form) and water-loaded purely siliceous zeolites (see Supplementary Table 1). We also added a two-dimensional silica bilayer (12 Å vacuum layer) used in ref. 38 with three different water loadings to the initial structure set. This bilayer resembles silicatene, a two-dimensional double-six-ring layer[81], but contains four-, five-, six-, and ten-rings (see Supplementary Table 1). All 153 initial configurations were then optimized under zero pressure conditions.

Next, the entire structure set was equilibrated for 10 ps at 1200, 2400, and 3600 K using AIMD simulations to sample reactive events at higher energies. Sampling of the low-energy parts of the PES used 210 unit cell deformations applied to all optimized structures (see Supplementary methods). Apart from microporous structures and two-dimensional BAS zeolites, we also added the same set of 210 lattice deformations for six dense BAS zeolite polymorphs, namely, four alumina polymorphs $\alpha$-Al$_2$O$_3$ (corundum)[82], $\theta$-Al$_2$O$_3$[83], $\gamma$-AlO(OH) (Boehmite)[84], and $\alpha$-Al(OH)$_3$ (Gibbsite)[85], as well as two aluminosilicate polymorphs, Si$_3$Al$_2$O$_{12}$H$_3$ (H$_3$O-Natrolite)[86] and Al$_2$Si$_2$O$_5$(OH)$_4$ (Dickite)[87]. Additionally, we (SOAP-FPS) subsampled AIMD trajectories of zeolite CHA taken from previous publications[15,45] to further extend the structure database. These trajectories are equilibrium MD runs of non-Löwenstein pairs (Al-O-Al) with various water loadings (0, 1, 15 water molecules)[45] and biased AIMD runs of Si-O(H) and Al-O(H) bond cleavage mechanisms[15].

For interpolation of the interactions in pure water, we performed AIMD simulations (10 ps) for bulk water with 64 water molecules at three densities (0.9, 1.0, 1.1 g cm$^{-3}$) and at three temperatures (300, 600, and 900 K). In addition, we used single water and water clusters *in vacuo* taken from the BEGDB database[50] (38 isomers from (H$_2$O)$_2$ to (H$_2$O)$_{10}$, available under: begdb.org) and four isomers of (H$_2$O)$_{20}$[88]. All clusters were first optimized (constant volume conditions) with a unit cell ensuring a distance between equivalent periodic images of at least 1 nm. Then the aforementioned 210 lattice deformations were applied to all optimized clusters. Finally, the unit cells of two ice polymorphs (Ice II[89] and Ice I$_h$[90]) were deformed in the same way for sampling of low-energy structures of crystalline water.

All AIMD simulations and structure optimizations were performed at the computationally less demanding PBE + D3(BJ)[35] level employing the dispersion correction of Grimme et al. (D3)[36] along with Becke-Johnson (BJ)[37] damping. The AIMD equilibration used the canonical (NVT) ensemble along with a 1 fs time step, the Nosé-Hoover thermostat[91,92], and with hydrogen being replaced by tritium. Structurally diverse configurations were extracted from the MD trajectories using SOAP-FPS (see Supplementary methods). These decorrelated MD structures were used, together with the generated set of lattice deformations, for single-point (SP) calculations at the (metaGGA) SCAN-D3(BJ) level[46]. The resulting SCAN+D3(BJ) reference dataset contained 248,439 structures.

An ensemble of six SchNet[51] NNPs was trained on the final SCAN +D3(BJ) database. The six independent training runs used different, randomly split parts of the DFT dataset with approximately 80% of the data points as training set and 10% as validation and test set, respectively. We used the same SchNet hyperparameters (6 Å cutoff, 6 interaction blocks, 128 feature vector elements, 60 Gaussians for distance expansion) and the mean squared error of energies and forces as loss function with trade-off 0.01 (high weight on force errors) as in our previous publication[38]. Minimization of the loss function used mini-batch gradient descent along with the ADAM optimizer[93] and four structures per batch. If the loss function for the validation set did not decrease in three subsequent epochs, the learning rate was lowered (from $10^{-4}$ to $3 \cdot 10^{-6}$) by factor 0.75.

### Generalization tests and reaction path searches

Testing of the NNP accuracy, robustness, and generality used a series of MD and NEB calculations of systems that were not included in the training database. To test the NNP performance at close-to-equilibrium (low temperature) conditions, we performed four MD runs (1 ns, 300 K) for zeolite FAU (primitive unit cell, 48 T-sites) at different chemical compositions. Three of the MD runs used a single Al atom (and BAS) per unit cell (Si/Al = 47) and three water loadings (1, 4, 48 water molecules). The fourth run was performed with 24 Al per unit cell (Si/Al = 1) and 48 water molecules. From every MD trajectory, 500 configurations were selected for subsequent SCAN+D3(BJ) and ReaxFF SP calculations. As an "inverse" test case to three-dimensional zeolites, we chose silicic acid Si(OH)$_4$ and aluminum hydroxide Al(OH)$_3$ solvated in bulk water (96 water molecules). Both systems were equilibrated at hydrothermal conditions 500 K for 1 ns. Subsequently, two hundred configurations were selected from both MD runs for accuracy evaluation.

To check the NNP performance and robustness for the sampling of reactive events, we first constructed a model of the external MFI-water interface. The starting point was an orthorhombic (96 T-site, taken from the IZA database) MFI unit cell with one silanol nest at T-site T9 for exploratory, high-temperature MD runs to sample reactive events at the internal and external MFI-water interface. Next, an MFI(010) surface model was created by adding a 12 Å vacuum layer and cleaving the Si–O bonds between the T-sites T7, T9, T10, and T12 (lattice plane with the lowest number of bridging O) yielding eight silanol groups on both surfaces. The resulting surface model is similar to previously used model systems[94,95] of synthesized MFI nanosheets[65]. After the addition of 165 water molecules, the model was equilibrated for 2 ns at 1600 K. One hundred structures were selected from the trajectory that include the surface defect creation shown in Fig. 2 for SP calculations. As an extreme case to test the NNP robustness at very high temperatures, we simulated the liquid state-of a BAS-water model system at 3000 K. The initial configuration was a model of GIS (32 T-site unit cells) with Si/Al = 1 and 24 water molecules which was equilibrated for 2 ns. Finally, two hundred structures were extracted from the MD run for the NNP and ReaxFF error evaluation.

For the FAU(Si/Al = 1) + 48H$_2$O system, we also performed an AIMD simulation directly at the SCAN+D3(BJ) level, in which we used the forces from the reference level of theory to propagate the structure in the MD simulation and then calculated the single-point NNP energies and forces for the selected snapshots from the AIMD trajectory. As the initial structure, we took an optimized structure from the FAU(Si/Al = 1) + 48H$_2$O test case simulations (see above) since the "pinned hydroxonium" species (see "Results" section) were observed to form during the NNP trajectory (see Supplementary Fig. 6). The NNP optimized initial structure was equilibrated for 10 ps at 300 K.

All MD simulations used a time step of 0.5 fs with hydrogen being replaced by deuterium, employing the Nosé-Hoover thermostat with a relaxation time of 40 fs[91,92]. The final generalization test set collected from all trajectories contains 2700 configurations for SP calculations at the SCAN+D3(BJ) and ReaxFF[58] level allowing the energy and force error evaluation shown in Table 1 and Fig. 1c (and Supplementary Table 3).

In addition, we optimized a set of seven silica structures (three polymorphs: $\alpha$-quartz, $\alpha$-cristobalite, tridymite; four zeolites: AFI, FER, IFR, MTW) at the NNP level (including the unit cell) for which experimental data (structures and enthalpies)[96–98] and DFT data (PBE+D3(BJ), SCAN+D3(BJ))[38] are available (see Supplementary Fig. 1 and Supplementary Table 4). This set of structures was chosen to check whether the current NNPs have the same accuracy as our previously developed NNPs for silica[38] and to evaluate how much the NNP energy/force errors change the structure and energetics of the DFT-optimized zeolites. Another accuracy comparison between the current NNPs and the previously developed silica NNPs involved a SCAN+D3(BJ) dataset of 1000 hypothetical zeolites randomly chosen from the hypothetical zeolite database[38] (see Supplementary Fig. 1 and Supplementary discussion). Here, we calculated energy and forces for this test set at the NNP level while the DFT data was taken from Ref. 38. We also calculated the adsorption energies of a single water molecule in MFI(Si/Al = 95) on all twelve (T1-T12) symmetry inequivalent T-sites and (randomly) chosen BAS (see Supplementary Fig. 7). The manually constructed initial structures were optimized at the NNP level (at constant volume). These MFI structures were then re-optimized at the PBE+D3(BJ) and SCAN+D3(BJ) levels to test the influence of NNP energy/force errors on structure optimizations and adsorption energy calculations in direct comparison to DFT.

Next, we conducted NNP performance tests for modeling of reaction pathways in FAU (primitive unit cell, Si/Al = 47) using NNP level climbing image NEB calculations. We chose four reaction pathways: (i) a proton transfer with one water molecule and without water (between O1–O4)[48,71,72], and (ii) water-assisted bond-breaking mechanisms of the Si–O2 and Al–O2(H) bonds[14,66,73] (see Fig. 3 and Supplementary Fig. 9). In addition, we tested a water-free proton transfer in CHA between O2 and O3. The numbering of the symmetry inequivalent oxygen atoms (see Supplementary Fig. 13) is consistent with the labeling of the zeolite frameworks in the IZA database (available under: iza-structure.org/databases). Energies at the SCAN+D3(BJ) were then obtained by SP calculations for all NEB images.

Lastly, the standard metadynamics[75] calculations both at the SCAN+D3(BJ) (using VASP package - see below) and the NNP (using PLUMED package - see below) level were used to evaluate the free energy profile of the proton jump in water-free CHA model with Si/Al = 11 (12 T-site unit cell), with the proton transferring between O2 and O3 framework oxygens. The collective variable defined as a difference of O2–H and O3–H distances was used. The simulations were run at 300 K, with the time step of 0.5 fs, the height of Gaussians set to 2.7 kJ mol$^{-1}$, and with the hydrogen replaced by tritium. The deposition rate and the Gaussian width were set to 50 and 0.02 Å for DFT calculations. The DFT calculation took about 60 ps until first recrossing. For NNP calculations, we could allow for longer simulations times (approx. 110 ps in total until recrossing), and thus slower deposition rate of 200 with the Gaussian width set to 0.04 Å was used (see Supplementary Fig. 12 for the effect of free energy parameters on NNP the free energy profiles).

## Δ-learning

We applied the Δ-learning approach[33] to improve the accuracy of our baseline SCAN+D3(BJ) model to the (hybrid DFT) $\omega$B97X-D3(BJ) level. First, we generated a training set using a subset of an (NNP level) biased dynamics run of a proton jump in CHA between O2 and O3, taken from Ref. 34. These structures were selected by FPS using the Euclidean distance of the (baseline) SchNet NNP representation vectors averaged over each MD snapshot. SP calculations were then applied to 500 extracted configurations to obtain energies and forces at the $\omega$B97X-D3(BJ) level.

The ΔNNP correction of the atomic energies $\Delta E_i$ to the NNP baseline model (SCAN+D3(BJ)) was obtained by linear regression of the SchNet representation vectors $\mathbf{x}_i$ of each atom $i$ with the (column) weight vector $\mathbf{w}_i$ and bias $b_i$: $\Delta E_i = \mathbf{w}^T\mathbf{x}_i + b_i$. We tested the ΔNNP model performance using 250 randomly chosen structures from the dataset and convergence tests showed that 150 training points give sufficiently low test set errors (see Supplementary Fig. 10). To test the ΔNNP quality, we repeated the NEB calculations described above for the water-free proton jump and Al–O2(H) bond hydrolysis in FAU as well as the proton transfer in CHA (O2–O3) without water. In addition, we performed two hundred SP calculations for structures taken from the biased dynamics runs of the proton jump (O1–O4) and the water-assisted Al–O2(H) bond cleavage to improve the force error statistics of the ΔNNP model.

## Biased dynamics using ML collective variables

Collective variables guiding proton transfer (O1–O4) and Al–O2(H) bond dissociation reactions in FAU described in the manuscript were trained using a variational autoencoder build on top of NNP-generated representation vectors.

To train the proton jump reaction, we used 3500 data points simulated from equilibrium MD runs on reactants and the same number on products. The data generated by using intuitively chosen CV and steered dynamics method were available for verification (see Supplementary Fig. 14) but were not used during training. The NNP representations were calculated and saved in a cache and an encoder producing a single CV was trained together with the decoder. We trained on 40 epochs with a learning rate of $10^{-4}$. The encoder generating the CV was a simple linear layer on top of pre-trained representations. To reduce the complexity of the task, only the 30 representation elements were chosen that maximized the variance between reactant and product configurations. For the biased dynamics, we used well-tempered metadynamics from the PLUMED package[75,99]. The parameters for the simulation are in Supplementary Table 8. The simulation was run for 1,800,000 steps using a 0.5 fs time step.

Al–O(H) bond dissociation was trained similarly to the proton jump case. We used the same number of data points obtained by running unbiased trajectories in both end states. The encoder was again only linear and was trained for 60 epochs with a learning rate $2 \cdot 10^{-4}$. 200 representation elements were pre-selected from the representation vectors generated by the NNP.

Parameters for the well-tempered metadynamics method are reported in Supplementary Table 7. We ran the biased dynamics for both test reaction for 1,500,000 timesteps of 0.5 fs. To improve the sampling of the desired reaction mechanisms we restricted the dynamics of some of the degrees of freedom. In particular, we required the distance between the free water molecule and the aluminum atom to be at most 2.2 Å to avoid it diffusing away. We also fixed two hydrogen atoms to their corresponding oxygens to avoid permutations that would complicate the process. In the same fashion, we disallowed the formation of the hydrogen bond (Fig. 3e-FS) for different O and H pairs. These restraints represent an approximation that may lead to a bias in the reported free energy profiles, however, the aim herein was to showcase the capabilities of the method rather than to obtain as accurate free energy profiles as possible. A detailed description of the autoencoder architecture, python code, and workflow can be found in ref. 34.

## Computational details

All simulations at the DFT level used the Vienna Ab initio Simulation Package[100–103] (VASP, version 5.4.4) along with standard versions of the Projector Augmented-Wave (PAW) potentials[104,105]. Calculations at constant volume were performed with an energy cutoff of 400 eV. Structure optimizations at constant (zero) pressure employed a larger energy cutoff of 800 eV. The minimum linear density of the k-point grids was set to 0.1 Å$^{-1}$ along the reciprocal lattice vectors. Single-point calculations using ReaxFF[58] were performed with GULP[106]. NNP training

and NNP level simulations used the Python packages SchNetPack[107] (version 1.0) and the atomic simulation environment (ASE)[108]. All initial structures were constructed using the Materials Studio suite along with its Solvate module to add water molecules to the unit cell[109]. Unless stated otherwise, the initial zeolite structure models for all simulation were taken from the IZA database[96,97] and then optimized in their siliceous form under constant (zero) pressure conditions. The subsequent dataset generation and test simulations used constant volume conditions.

## Data availability
The trained Neural Network Potentials (NNP and ΔNNP model), scripts, and all energy and force data used in this work at the (Δ)NNP, ReaxFF, and DFT (SCAN+D3(BJ) and ωB97X-D3(BJ)) level as well as the DFT (SCAN+D3(BJ)) training data are publicly available in a Zenodo repository under CC-BY-NC-SA 4.0 license: https://doi.org/10.5281/zenodo.10361794.

## Code availability
The python package enabling automatic generation of the collective variables using variational autoencoders (see also ref. 34) including examples with data is available at https://doi.org/10.5281/zenodo.10938030 and the modified PLUMED library enabling use of these ML-based collective variables with PLUMED can be found at 10.5281/zenodo.10938033.

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

## Acknowledgements

Charles University Centre of Advanced Materials (CUCAM) (OP VVV Excellent Research Teams, project number CZ.02.1.01/0.0/0.0/15_003/0000417) is acknowledged. L.G. acknowledges the support of the Primus Research Program of Charles University (PRIMUS/20/SCI/004) and that of Czech Science Foundation (23-07616S). Computational resources were provided by the e-INFRA CZ project (ID:90254), supported by the Ministry of Education, Youth and Sports of the Czech Republic. C.J.H. and L.G. acknowledge support from the Charles University Centre of Excellence award UNCE/SCI/014.

## Author contributions

A.E. performed the simulations needed to obtain the dataset, curated the training/testing dataset, e.g., using active-learning strategies, trained and validated the NNP models (both baseline and Δ-learned), conceived and carried out the bulk of the generalization tests; analyzed the data, wrote the original manuscript draft and contributed to its later refinement. M.Š. generated the collective variables using the baseline NNP, carried out the metadynamics simulations, and co-wrote the sections on accelerating rare event sampling using baseline model representations. I.S. carried out a part of generalization tests, in particular the transition state modeling, and structure analysis of the silica database. P.N. acquired funding and partially supervised the work. C.J.H. partially supervised the work, provided some initial datasets, co-wrote and revised the manuscript. L.G. acquired funding, supervised the work, contributed to data analysis/curation, carried out and analyzed the metadynamics simulation of proton jump in H-CHA, conceived the extensions of the baseline model, and co-wrote and revised the manuscript.

## Competing interests

The authors declare no competing interests.
