## [Peer Review File · Nature Communications]

A reactive neural network framework for water-loaded acidic zeolitesREVIEWER COMMENTS

Reviewer #1 (Remarks to the Author):

This is, generally speaking, an interesting manuscript on a very topical area of research, soundly written. The promise made in the title and abstract is very broad, and in my view, overly broad and general, without data demonstrating how it is fully achieved. In particular, while the approaches chosen in general follow the state of the art, the testing of the model and the demonstration of its statistical properties (and the learning itself) are not fully available in the text presently. Also, the supporting information is very succinct, and the code and data are not fully available. I do not think the work is fully reproducible by others in the field, in the current version of the manuscript and associated files and repositories.

Finally, regarding the criterion of novelty and editorial interest: I am not sure if Nature Communications is the best journal for this work, which is quite technical. Indeed, it is a follow-up and extension of some of the authors' previous study in <https://www.nature.com/articles/s41524-022-00865-w>. I feel that the authors wanted to restrict themselves in the manuscript for a broad audience journal, but the drawback is that now the writing is somewhat superficial and many interesting discussions on the methodology are avoided. Going in detail:

One first question, which I do not think is sufficiently considered in the manuscript, is the nature of the learning process/training, and the quantity of data necessary to produce the model. I was very surprised to find (unless I missed them) no learning curves, no plot of the dependency of the validation error as a function of the training set size. Is the model trained still in the data-limited regime, or not? This is a crucial question, as 248 439 structures is a large data set compared to other studies in the field: is the approach chosen here less data efficient? Or could fewer structures have been used? The authors do not discuss data efficiency at all (except in the Delta-learning section).

Regarding the choice of hyperparameters, I am surprised that the authors took the same values as in their previous work, without testing other possible choices, and discussing the importance (or not) of the parameters that have a physical meaning. In particular, I am surprised at the choice of a 6 angstrom cutoff value, especially as the authors discuss processes that directly involve longer distances, and chemical reactivity. Have they studied the influence of this crucial parameter? Why do they not expect a larger cutoff to make more chemical sense?

The authors compare their work with ReaxFF, which is a notoriously bad reactive force field of zeolites. I understand their point in highlighting the quality of the model they produce, but choosing such a low-quality reference point seems like a low bar to set. Have the authors tested, on specific structures (all-silica zeolites for example), this force field against models previously published, including theirs? In gaining accuracy on a broader chemical range, does this deteriorate the quality of the description of all-silica systems?

The data and code provided are not sufficient for a full reproduction of the work performed by the authors. In order to meet best-effort guidelines used by other groups, the authors should also publish: 1. the full training and testing data, both for the reference model and the delta-learning study, 2. the input files or codes used to perform the MD simulations. The authors state of this "The remaining data for the reproduction of results is available upon reasonable request." but that is frankly insufficient to meet the standards of the field, and the policies of Nature Communications as I read them.

Reviewer #2 (Remarks to the Author):

This manuscript is relevant for the zeolite and nanoporous materials research community. The paper is well organized and includes great detail about the tests and validations carried out. The authors provide relevant references, although this reviewer considers that there are a couple of recent publications missed.

<https://pubs.acs.org/doi/10.1021/acs.jpcc.2c08429>

<https://pubs.rsc.org/en/content/articlelanding/2022/SC/D2SC01225A>

The claims and conclusions are well discussed. The datasets in the supplementary files do not include any instruction to open them, therefore is nearly impossible to use and verify the models generated. Similar to the files in the repository Zenodo, where one has to request access, the model files are only accessible upon request.

The text has some inconsistencies in the nomenclature, for example: "metaGGA" and "(meta)GGA".

The statement "One of the challenges in training general NNPs for a material class is creating an interpolation grid that captures relevant parts of the configuration and chemical space." is confusing. It is not clear what is an "interpolation grid" in this context, and whether they mean that training creates a grid. There is no explanation of how to calculate the "energy error distribution", ΔE_r , and what is the reference or whether the reference is the same every time this term is used. The authors should mention the relevance of the quantum effects of protons in MD, and how this could affect so conclusions.

In this reviewer's opinion, the Delta-learning section is not relevant and does not contribute to the results. The authors claim that the Delta-learning approach could help to get better results using two exchange-correlation functional, lower level and higher level, in this case, hybrid-GGA and metaGGA. However, the NNP in this paper was trained over a metaGGA reference, and hybrid was only used for the Delta-learning portion, therefore they go from higher to lower level, instead going in the opposite direction and improving the metaGGA. Also, there is no evidence that the Delta approach constructed here is computationally less demanding.

This work could be published after the authors address previous comments and improve the documentation of the supporting materials.

Reviewer #3 (Remarks to the Author):

The manuscript is situated in a very interesting research field, namely the derivation of machine learning potentials for reactive events in zeolites in presence of water. A huge amount of work has been done by the authors and the results are promising. However in its current form, the manuscript combines a lot of aspects and the authors give a series of bold statements on the final goal of the research which is not fully substantiated by a proper validation of the neural network potential. Furthermore a rather extensive comparison is performed with ReaxFF. In the opinion of this reviewer, the parts on ReaxFF should be drastically reduced or even be fully removed from the main manuscript. ReaxFF is not a good comparative basis for neural network potentials as the latter have the goal to obtain results at quantum mechanical accuracy. Within the zeolite community ReaxFF is not often used, as it depends on a lot of parameters and can not be regarded as chemical accurate. In summary the manuscript in its current form, is designed to show that a lot can be done, without real validation. Furthermore the authors claim that in the current manuscript, a toolset is built to describe in a more general way reactive events in zeolites in presence of water, but it is unclear how the authors see the usage of the methodology? Is the derived neural network potential transferable in the sense that it can directly be used by other researchers working in the field of zeolite catalysis or is it the intention that other researchers working in the field of zeolite chemistry, derive their own potential based on the methods presented in the paper? If the final goal is the first one, it would be necessary that the potential is made fully available to the research community, together with necessary scripts to use the potential. There are some concerns with respect to the data availability (see comments later in this report). It is thus not fully clear which message the authors finally want to bring to the community. In the opinion of this reviewer, it would be better to bring a more focused message with data that are thoroughly validated.

More detailed comments are given below :

Abstract

- The abbreviation H-AS is not widely used within the zeolite field.
- The sentence “we have developed a general potential surface interpolator with consistent accuracy” should be revised. It would be better to state that a machine learning potential for the Potential Energy Surface has been derived instead of using the terminology interpolator. It is not clear what is meant with consistent accuracy. The terminology chemical accuracy is used in the field to refer to methods where the errors on the energies are less than 1kcal/mol, one could also state that an accuracy the same as the underlying training data for example obtained at DFT level is obtained, but consistent accuracy is not the right terminology in this context.
- “Furthermore, we show that the baseline model.... collective variable”. This sentence is confusing and showcases the divergent focus of the article.
- “This framework allows for operando simulations ...” This statement is much too bold and should be toned down. The manuscript does not provide all tools to simulate realistic catalysts at operando conditions. Furthermore the terminology quantitative accuracy is again confusing, see the earlier remark on the definition of accuracy.

Introduction

- Page 3 : Last sentence and Page 4 first sentence. Very bold statements are made, the authors claim to cover all aluminosilicates, at all possible water compositions, aluminum substitutions and reactive events, thus they claim to develop a potential which is able to describe almost the whole field of zeolite catalysis, with the exception of metal-loaded zeolites. The results presented in the current paper, indeed show that the authors can simulate a lot of things, but in the opinion of this reviewer, the work does not show that the whole field of Brønsted acidic zeolites can be simulated with the work proposed in this manuscript.

Results

Database generation and training of the general H-AS NNPs

- The motivation for the structures to be taken up in the training set is not clear. A rationale should be given why certain zeolite models were taken into account in the training data. The authors used three existing zeolites namely SOD, CHA and MVY and 7 hypothetical zeolites. Was it the intention to cover the whole zeolite atlas? Why did they choose SOD, CHA and MVY as existing frameworks? Is there any rationale for the structures in terms of the secondary building blocks needed to be present to account for the whole set of materials? More explanation should also be given on the hypothetical zeolites, which secondary building blocks do they contain? This motivation and clarification should be taken up in the main manuscript.
- The authors took also slab systems into account, it is unclear how these were constructed and terminated. How do these models compare with slab models that were modeled by Chizallet and co-workers or with models taken up in the database of Sholl (10.1021/acs.chemmater.8b03290).
- It seems that the simulations to generate the training data set were performed at very high temperatures from 1200K to 3600K, which is much higher than any catalytic process. Some catalytic processes take place at high temperatures e.g. 700-800 K. It is motivated that high temperatures are necessary to sample reactive events, but in how far are the materials still stable at these conditions?
- Varying Si/Al ratios are considered. Also here a motivation should be given of where Aluminum was placed and why, the number of symmetrically equivalent Aluminum sites, the proximity of Aluminum etc. Aluminum siting is a very challenging topic in the field of zeolite catalysis, it would be perfectly understandable if the authors would not be able to cover the whole range of Si/Al ratios, but in the current form of the manuscript it is claimed that they cover the whole space of Si/Al ratios, therefore it is necessary to address these issues, otherwise they need to tone down the statements made in the manuscript.
- Is it correct that no explicit enhanced sampling simulations were done to generate information for the training of the neural network potential? As the authors claim to cover the whole space of reactive events, were enough reactive events taken up by performing the superhigh temperature simulations? In this case the authors should comment more on the reactive events they observed during the high temperature simulations. In the current version of the manuscript and supporting information nothing is mentioned about these aspects.
- How much training data were eventually used for the generation of the neural network potential?
- Why did they authors use SchNet architecture? Much more efficient mathematical frameworks are available, which allow to reduce the number of training data substantially and also allow to reduce the errors.
- The t-SNE plot is instructive, however the authors should comment in how far they are able to describe diverse topologies of the zeolite database and currently in the figure only FAU, MFI, etc is indicated. Is the whole zeolite database well described?
- In terms of validation, the authors refer to Table 2 in the Supplementary information. More information should be provided in the construction of the training and validation set.

Generalization tests and exploration of configuration space

- It is suggested to remove the comparison with ReaxFF completely from the main manuscript and focus on the performance of the NN potential with respect to the underlying DFT data. It is obvious that that neural network potential is much better than the ReaxFF, this can be mentioned once and if necessary the results on ReaxFF can be taken up in the SI.
- Table 1 : How were the validation systems generated? For example FAU+48 H₂O molecules, how were the structures of this systems determined? From MD simulations? If yes, how long were the simulations? A lot of details are missing for the validation protocol.
- Table 1 : it is unclear what the reference was for the errors. Are the methods validated with respect to underlying DFT data?
- It is suggested to completely remove the rather extensive discussion on ReaxFF in the main manuscript.

Sampling of equilibrium properties

- Reference is made to [46] a paper on chemrxiv (<https://doi.org/10.26434/chemrxiv-2022-d1sj9>), which does not seem to exist.
- Page 10 : simulations on FAU with various Si/Al ratio and water loadings. From the discussion it is unclear which simulations are performed with the NNP and DFT? Are DFT simulations done for validation? For example an adsorption energy of -79 kJ/mol is reported, probably this value was obtained with the NNP? Was the DFT adsorption energy also calculated? The authors refer to reference 45, which suggests that the DFT simulation was not performed? The same remarks hold for the further discussion, where the degree of solvation of the BAS is discussed. Qualitative features are discussed on the degree of solvation obtained with the NNP, but it is not clear in how far these observations correspond to simulations performed at the DFT level. It only shows that the NNP is stable during the simulations.
- Page 11 : “In both cases, the NNPs accurately reproduce the SCAN+D3(BJ) energies and forces.” This sentence seems to suggest that DFT validation runs were performed. Can the authors clarify which simulations have been done in production with the NNP and which validation runs have been performed?

NNP performance for highly activated reactive events

- Simulations are performed at extreme conditions, namely 1600K and 3000 K, to force reactive events. Such simulations are not very realistic. The authors do observe some reactive events like the formation of silanol defects etc. These simulations prove that the system remains stable under these extreme conditions but there is no validation that the NNP yields realistic chemical processes. On the positive side, the authors do validate the energies and forces for a series of snapshots during these simulations and observe that the errors remain within acceptable range. However to truly validate the NNP it would be necessary to simulate some free energy barriers for some chemical processes both with the NNP and the underlying DFT level of theory. It would be necessary to validate that the DFT methodology and the NNP samples similar parts of phase space and obtains similar free energy barriers for the activated processes.

Extensions of the NNP model

- The delta learning procedure is promising, however more details should be given on the methodology.

- Figure 3 : It seems that the errors made by the delta learning procedure are of the same order of magnitude than the delta between the levels of theory. This needs to be clarified and it needs to be validated in how far the methodology would work for levels of theory which are further apart.
- The part on the acceleration of the rare events is interesting but confirms the general statement that a bit of everything is collected in this paper without real focus. It is also questioned in how far the procedures are made available for the rest of community. If it is the intention to show that a lot can be done by the methods described in the paper, it would be necessary to provide the necessary tools for the rest of the community to actually use the methods.

Conclusions

- I reiterate some of the statements formulated above, very bold statements are made, namely that the tools will enable to simulate hydrothermal instabilities, formation of defects, etc under operating conditions. It is not clear which tools are made available for the community and in how far these are sufficiently validated to be used in a more general context.

Data availability

It is unclear which data are made available. It seems that the potential is not provided in the zenodo database? Following statement is mentioned on the zenodo link : Trained Neural Network Potentials NNPs including the Δ -learned model (SchNetPack, version 1.0) are available under the CC BY-NC-SA 4.0 license for non-commercial use only. Furthermore following statement is given You may request access to the files in this upload, provided that you fulfil the conditions below. The decision whether to grant/deny access is solely under the responsibility of the record owner. It is thus unclear whether the potential is made available or not? Related to this point, reference is made to the earlier remark in this report, namely what is the final intention of the manuscript? Providing a potential that can be used by the research community or showcasing a methodology to be used by other researchers? Irrespective of the final goal, it is necessary that all information including the NNP, the training data, the validation set and scripts are made available for the research community.

Reviewer #4 (Remarks to the Author):

In the submitted manuscript, a reactive neural network potential (NNP) reproducing the reference metaGGA DFT level is proposed for describing zeolite systems, possibly loaded with water. The utility of the potential is demonstrated on a number of examples that involve the sampling of equilibrium states (water adsorption in faujasite and chabazite) and description of activated processes (surface defects formation) in static and MD calculations. Furthermore, the use of Δ -machine learning for accuracy improvement (via changing the electronic structure method) is proposed and demonstrated on static and metadynamics calculations of reactions of proton transfer and Al-OH bond dissociation in faujasite. The manuscript is well written, the scientific presentation is sound, and the practical examples presented provide a clear demonstration of the usefulness of the approach. I recommend its publication in Nature Communications provided the comments listed below are carefully addressed.

- 1.) According to data presented in the paper and SI, the mean absolute error achieved for forces is of order of 0.1 eV/\AA . I wonder if this is sufficient accuracy for the applications discussed in the manuscript (relaxations and MD simulations). In relaxations, for instance, the relaxation criterion for forces used in practice is of order of 0.01 eV/\AA or less and, clearly, a noise that is an order of magnitude greater than this is likely to confuse the relaxation algorithm or even to yield structures that are significantly different from the structures relaxed without the use of machine learning potentials. The authors should clearly explain what their requirements for accuracy of the NNP are and what motivated their choice.
- 2.) Related to the previous point - it is not clear from the presented data (e.g., Supp. Tab. 2) to which extent the error in forces/energies correlates with the forces/energies from explicit DFT calculations. This is very important, because, if the error in forces is not random (i.e., uncorrelated to the actual force), the NNP will bias the sampling of configuration space. To this end, a careful covariance analysis would be useful.
- 3.) The data presented in Tab. 3 compare the results obtained using NNP and Δ NNP with the explicit DFT calculations for the Al-OH bond dissociation in FAU. Given the data are meant to demonstrate how the NNP can be used for "Improving the baseline model", the numerical results are, in my opinion, rather unconvincing, because the error in the relative positioning of stationary states on PES are as large as the differences between the two electronic structure methods discussed. In other words, the SCAN+D3-based NNP results seem to be as good/bad proxy to the explicit ω B97X-D3 calculations as the ω B97X-D3-based Δ NNP making the Δ NNP useless. For the state "TS1", for instance, the explicit ω B97X-D3 yields 85 kJ/mol , while the ω B97X-D3-based Δ NNP and the SCAN+D3-based NNP models predict 93 and 77 kJ/mol respectively, i.e., the error with respect to the explicit result is the same for both models. Even worse, the positioning

of the state label “I” obtained with the explicit ω B97X-D3 is 60 kJ/mol, the ω B97X-D3-based Δ NNP model overestimates this value by 8 kJ/mol, while value obtained from SCAN+D3-based NNP model is only 3 kJ/mol lower than the explicit ω B97X-D3 result. The authors should explain better why they think their Δ NNP procedure is useful.

- 4.) Page 27: *“To avoid simulating degrees of freedom that are unimportant for the Al–O(H) bond dissociation reaction we introduced some restraints to the biased dynamics. We required the distance between the free water molecule and the aluminium atom to be at most 2.2 Å to avoid it diffusing away. We also fixed two hydrogen atoms to their corresponding oxygens to avoid permutations that would complicate the process. In the same fashion we disallowed the formation of the hydrogen bond (Figure 3d-FS) for different O and H pairs.”* I disagree that the restrained degrees of freedom are unimportant for the Al-OH dissociation reaction. Clearly, restricting the distance between water and Al significantly affects the entropy contribution to free energy of the given state. In other words, “diffusing away” is a way how entropy stabilizes the system and, when avoided, the sampling is biased towards states that might be irrelevant at the given thermodynamic conditions.
- 5.) The data presented in Supp. Tab. 6 are not quite clear to me – what is the meaning of MAE and RMSE for reaction energy (E_r), which is a single value obtained either as potential energy difference between two minima (relaxations) or difference between ensemble averaged potential (or total) energies of two stable states? For such a quantity, NNP should yield a single number, and hence, trivially, the MAE and RMSE should be identical (equal to absolute value of difference between the E_r from explicit and NNP calculations). Likewise, the way the F and ΔE were obtained should be presented in more detail – the present version of SI is very scarce about that (just one sentence).
- 6.) Some of the items in Supp. Tab. 7 are not unitless but their units are not provided.

RESPONSE TO REVIEWERS' COMMENTS

We thank the Reviewers for their detailed suggestions and constructive criticism. We have thoroughly revised the manuscript (and Supplementary Information) following the Reviewers' suggestions, significantly extended the Supplementary Information with technical details and most importantly added four new validation tests that compare the performance of NNPs vs. the reference DFT (SCAN-D3(BJ)) side-by-side for geometry optimizations (and property predictions based on geometries), equilibrium MD simulations and generation of the free energy profiles.

Below, we have responded to each point (in black) made by the reviewers with our comments (blue). In addition, we have provided the manuscript file with the changes explicitly highlighted (olive).

Lastly, we would like to stress that we have now made the data, including both the potential(s) and the full training dataset, fully publicly available (under CC BY-NC-SA license at <https://doi.org/10.5281/zenodo.10361794>), which should make this work reproducible and allow for unhindered use by the whole community (the data from generalization test were provided already in the original submission). We note that this goes beyond the requirements of the data availability policy of Nature Communications, and we expect it to maximize the value of the work to the scientific community. The reason for us not to provide originally the potentials (and training dataset) is the fact that we believe that these data have a reasonably high commercial potential and we are concerned that an unrestricted (e.g., CC BY) license would have the capacity to jeopardize ongoing commercialization projects/collaborations. However, we now found a legally sound way to allow the academic community unrestricted, open access to this data while still restricting commercial use.

Reviewers' comments:

Reviewer #1 (Remarks to the Author):

This is, generally speaking, an interesting manuscript on a very topical area of research, soundly written. The promise made in the title and abstract is very broad, and in my view, overly broad and general, without data demonstrating how it is fully achieved. In particular, while the approaches chosen in general follow the state of the art, the testing of the model and the demonstration of its statistical properties (and the learning itself) are not fully available in the text presently. Also, the supporting information is very succinct, and the code and data are not fully available. I do not think the work is fully reproducible by others in the field, in the current version of the manuscript and associated files and repositories.

We thank the reviewer for a generally positive evaluation of our work and suggestions for its improvement. We have incorporated the suggestions made by the reviewer, namely:

1. We have added four new models to support the generality and robustness of our neural network potential (NNP)
 - a. The free energy calculation using "vanilla" metadynamics of proton jump in H-CHA model comparing performance of NNP to fully first principles biased simulation at the reference-level employing SCAN+D3 XC DFT functional.
 - b. The equilibrium molecular dynamics simulations (MD) at the SCAN+D3 for a novel "pinned hydroxonium" species observed in during NNP MD runs complemented by single point (SP) calculations at the NNP level. This is an out-of-domain (OOD) system.
 - c. The static water adsorption energies on all distinct T-sites (T1-T12) in H-MFI zeolite, i.e, in the framework that is an OOD system for the potential. Separate geometry optimizations done at NNP, SCAN+D3 and PBE+D3 levels.
 - d. Evaluation of multiple properties (energies, densities, average Si-O bond distances, lattice parameters) of the purely siliceous database used also in our

previous work (<https://doi.org/10.1038/s41524-022-00865-w>). Again it is an OOD test (see Supplementary Figure 1 and Supplementary Table 4).

2. We have significantly extended the supporting information, providing: data about the learning process (learning curves); hyperparameter benchmarking; covariance analysis of DFT vs NNP energies and forces; detailed description of the zeolite database used and other detailed information about additional models considered above (see point 1)
3. We have made both the potential(s) and the full dataset fully publicly available (under CC BY-NC-SA license) which together with other data provided should make this work fully reproducible.

Based on the similar suggestions of other reviewers, we have toned down the promise made in the abstract and throughout the manuscript (in Introduction and Conclusions sections in particular), instead focusing on the multiple challenging OOD (generalization) tests, which confirm the transferability of the NNP in these scenarios. While it is impossible to fully prove the generality of these, or any potentials across such a broad configurational and chemical space as considered in this work, we believe that the generalization tests are sufficiently diverse to claim an expectation of generality (for example, see the analysis in Fig 1b and the SI Figs 1-2).

Finally, regarding the criterion of novelty and editorial interest: I am not sure if Nature Communications is the best journal for this work, which is quite technical. Indeed, it is a follow-up and extension of some of the authors' previous study in <https://www.nature.com/articles/s41524-022-00865-w>. I feel that the authors wanted to restrict themselves in the manuscript for a broad audience journal, but the drawback is that now the writing is somewhat superficial and many interesting discussions on the methodology are avoided. Going in detail:

We have now added a significant amount of additional information to the supporting information, which should make our contribution relevant also for more technically minded readers, while keeping the level of detail high.

We also note that BAS zeolite systems, for which we have developed the NNPs, are industrially very relevant systems, making our work of interest for a broader audience. In addition, our contribution shows that nowadays one can develop **reliable reactive** machine learning potentials for a broad chemical and configuration space, i.e., use them to model **realistic** materials reactively.

We do not agree with the reviewer's opinion that current work is only an extension/follow-up on our previous study (<https://www.nature.com/articles/s41524-022-00865-w>). We would argue against this from two viewpoints:

1. The current work describes a development of the NNP that attempts to cover vastly broader configuration and chemical space. In our previous work we considered a simplified system of two elements. In this work, we consider a hugely diverse class of industrially relevant systems of great application importance, containing multiple phases (solid-liquid interfaces), four elements, and allowing for an arbitrary combination of molecular and crystalline species. *The ability to train such an NNP using only few hundred thousands of data points (and likely even less would suffice - see responses to other points below) is an important message to a general audience as it indicates an ability to break or at least move the "curse of dimensionality" to even broader chemical and configurational spaces.*
2. Herein we exemplify how these NNPs can be embedded in a general ML-based framework that can: i) improve the accuracy of the NNP for a reaction/process of interest using the delta-learning approach, and ii) help to discover reaction mechanisms and their low-dimensional representations (reaction coordinated, collective variables) using herein developed (intermediates) of NNPs. *This is an example of a "whole-solution-package" to study the chemistry of zeolites (but also other materials) in a rather reliable, (semi)-automatic, high-throughput way with*

tunable accuracy - again we would claim that this represents a step change over our previous work and a point of significant novelty.

One first question, which I do not think is sufficiently considered in the manuscript, is the nature of the learning process/training, and the quantity of data necessary to produce the model. I was very surprised to find (unless I missed them) no learning curves, no plot of the dependency of the validation error as a function of the training set size. Is the model trained still in the data-limited regime, or not? This is a crucial question, as 248 439 structures is a large data set compared to other studies in the field: is the approach chosen here less data efficient? Or could fewer structures have been used? The authors do not discuss data efficiency at all (except in the Delta-learning section).

It is true that the generated database is much larger than those used in studies that focussed on specific systems/reactions, e.g., those with fixed chemical composition (for example: <https://www.nature.com/articles/s41467-022-32294-0>). However, our DFT dataset is much smaller than databases (e.g., OC22, <https://doi.org/10.1021/acscatal.2c05426>) that aim to span the whole periodic table containing millions of DFT data points. Recently, SchNet NNPs were trained on one proton jump reaction in one single zeolite using several 100k structures (<https://www.nature.com/articles/s41467-023-36666-y>). Therefore, the use of ~250k configurations seems not an extraordinary database size to train vastly more general potentials that cover the chemical and configuration space across aluminosilicate zeolites with four elements and distinct solid-liquid interfaces.

We have also added Supplementary Figure 3 which shows test set energy/force errors as a function of the number of training structures. We found the lowest NNP energy and force errors for training set sizes of ~170k-200k configurations. Smaller training set sizes show only slightly larger errors demonstrating that the data set size is not a limiting factor of the NNP accuracy. Certainly, more data-efficient equivariant NNP architectures would probably require even less data points for comparable NNP performance. However, performance and generality, rather than minimizing database size was the primary focus of this work.

Regarding the choice of hyperparameters, I am surprised that the authors took the same values as in their previous work, without testing other possible choices, and discussing the importance (or not) of the parameters that have a physical meaning. In particular, I am surprised at the choice of a 6 angstrom cutoff value, especially as the authors discuss processes that directly involve longer distances, and chemical reactivity. Have they studied the influence of this crucial parameter? Why do they not expect a larger cutoff to make more chemical sense?

We used a well tested setup of hyperparameters (<https://www.nature.com/articles/s41467-023-36666-y#MOESM1>, <http://aip.scitation.org/doi/10.1063/1.5019779>). Thanks to the message-passing architecture of SchNet, the learned representation vectors for each atomic environment contains some information from atoms further away than the short-range distance cutoff, due to the use of several interaction blocks (six in our case). To illustrate the NNP accuracy as a function of these parameters, we have included the energy/force errors as a function of cutoff and interaction blocks to Supplementary Figure 3. Our employed setup (6 angstrom cutoff, 6 interaction blocks) was found to show the best performance, in line with previous studies (see above).

The authors compare their work with ReaxFF, which is a notoriously bad reactive force field of zeolites. I understand their point in highlighting the quality of the model they produce, but choosing such a low-quality reference point seems like a low bar to set. Have the authors tested, on specific structures (all-silica zeolites for example), this force field against models previously published, including theirs? In gaining accuracy on a broader chemical range, does this deteriorate the quality of the description of all-silica systems?

Unfortunately, ReaxFF is the only reactive force field that covers the BAS zeolite systems, i.e., making it the only relevant comparison. Therefore we focused on comparison with this, as the Reviewer mentioned, “low-bar” reference. However, we acknowledge the concerns of the Reviewer, which were also shared by other Reviewers and so we have significantly reduced the discussion of ReaxFF performance to a very brief discussion in the section “Generalization tests and exploration of configuration space” (including Figure 1c) and the rest of comparisons with ReaxFF have now been moved to supporting information (SI).

We thank the Reviewer for a suggestion to compare the performance of our NNP for BAS zeolite systems against other models for simple systems (such as all-silica zeolites) - we have carried out this test against our old silica-only NNP (<https://www.nature.com/articles/s41524-022-00865-w>) (see Supplementary Table 4 - **silica tests**) and found out that the quality of the description of new NNPs does not deteriorate for all-silica systems. This is another encouraging result for the herein-developed NNPs, as it shows that it is possible to increase the complexity of the database significantly without the loss of accuracy (compared to simpler systems/datasets), even using the current (rotationally invariant) NN architectures.

The data and code provided are not sufficient for a full reproduction of the work performed by the authors. In order to meet best-effort guidelines used by other groups, the authors should also publish: 1. the full training and testing data, both for the reference model and the delta-learning study, 2. the input files or codes used to perform the MD simulations. The authors state of this "The remaining data for the reproduction of results is available upon reasonable request." but that is frankly insufficient to meet the standards of the field, and the policies of Nature Communications as I read them.

We refer the reviewer to a general response about the data availability issues provided elsewhere in the response letter, e.g., in the general introductory part to the response letter above.

Reviewer #2 (Remarks to the Author):

This manuscript is relevant for the zeolite and nanoporous materials research community. The paper is well organized and includes great detail about the tests and validations carried out. The authors provide relevant references, although this reviewer considers that there are a couple of recent publications missed. <https://pubs.acs.org/doi/10.1021/acs.jpcc.2c08429>
<https://pubs.rsc.org/en/content/articlelanding/2022/SC/D2SC01225A>

We thank the reviewer for a very positive evaluation of our work. We have added the mentioned references to the manuscript.

The claims and conclusions are well discussed. The datasets in the supplementary files do not include any instruction to open them, therefore is nearly impossible to use and verify the models generated. Similar to the files in the repository Zenodo, where one has to request access, the model files are only accessible upon request.

We acknowledge the issue of availability of the data and therefore we have made the data, including both the potential(s) and the full dataset, fully publicly available (under CC BY-NC-SA license), which should make this work reproducible and allow for unhindered use by the whole community.

The text has some inconsistencies in the nomenclature, for example: "metaGGA" and "(meta)GGA".

The nomenclature has been unified to “metaGGA” throughout the manuscript.

The statement "One of the challenges in training general NNPs for a material class is creating an interpolation grid that captures relevant parts of the configuration and chemical space." is confusing. It is not clear what is an "interpolation grid" in this context, and whether they mean that training creates a grid.

The sentence mentioned by the reviewer has been rephrased to remove confusion and reads now: "One of the challenges in training general NNPs for a material class is creating a training database that captures relevant parts of the configuration and chemical space".

There is no explanation of how to calculate the "energy error distribution", ΔE_r , and what is the reference or whether the reference is the same every time this term is used.

We used two energy error metrics in this work to evaluate the accuracy of the (Δ)NNP models with respect to their DFT reference. First, a reaction energy error ΔE_r (of a formation reaction in Eq 1 of the original manuscript) to compare test systems across the entire chemical BAS-zeolite space. Therefore, ΔE_r uses the same reference structures for all presented test cases (i.e., quartz, alumina and gas-phase water) quantifying the NNP quality for transformations that involve a change of chemical composition (adsorption, reaction or cohesive energies). Secondly, we used the error $\Delta \Delta E$ of the relative energies ΔE with respect to a reference structure with the same chemical composition, e.g., the initial structure of an MD trajectory or NEB calculation, i.e., using different reference structures for each test case quantifying NNP (ReaxFF) errors for this chemical composition or better a specific thermodynamic state point (see, e.g., <https://www.pnas.org/doi/abs/10.1073/pnas.2110077118>) only.

We have added a more detailed description of the two energy error metrics in the Results part and to the Supplementary Information to clarify the definition of ΔE_r and ΔE .

The authors should mention the relevance of the quantum effects of protons in MD, and how this could affect so conclusions.

We have added a brief discussion of the relevance of nuclear quantum effects (NQEs) to the "NNP performance for highly activated reactive events" part mostly relating to a recent work from Bocus et al. (<https://doi.org/10.1038/s41467-023-36666-y>) who did focus on this issue and quantified that inclusion of NQEs can lead to lowering of the proton jump barriers by 5-10 kJ/mol depending on the temperature. Otherwise, it is out of scope of the current contribution to quantify the effect of NQEs in general. Nevertheless, our NNPs (similarly as done by Bocus et al.) represent a tool that can be straightforwardly deployed to evaluate NQEs within path-integral molecular dynamics simulations and since our potentials sample much broader configurational space than those of Bocus et al., we expect them to be better suitable for NQE simulations than MLPs trained only for specific reactions/systems.

In this reviewer's opinion, the Delta-learning section is not relevant and does not contribute to the results. The authors claim that the Delta-learning approach could help to get better results using two exchange-correlation functional, lower level and higher level, in this case, hybrid-GGA and metaGGA. However, the NNP in this paper was trained over a metaGGA reference, and hybrid was only used for the Delta-learning portion, therefore they go from higher to lower level, instead going in the opposite direction and improving the metaGGA. Also, there is no evidence that the Delta approach constructed here is computationally less demanding.

The subsection "Improving the baseline model using delta-learning" has been thoroughly revised following also the comments from the other reviewers regarding this part. In particular, we highlighted much more clearly the great performance of the delta-learning model for the in-domain case, i.e., proton jump in H-CHA model, which is a common way to evaluate performance of the delta-learning model. Only then did we focus on much more challenging cases, in which we tested the ability of the delta-learning model to generalize to out-of-domain

cases such as proton jump in different framework (H-FAU) and a different reaction in different framework (Al-O(H) bond dissociation in H-FAU). In the responses to other reviewers' comments below we also show a more complex example of using delta-learning on top of our baseline NNP, which illustrates its capabilities in conjunction with the robust baseline NNP - however, we are not including this in the current manuscript as it is out of scope here and is part of an ongoing project.

To assess particularly the issues raised by the reviewer here, we have rephrased the initial paragraph of this subsection to clarify the procedure and purpose of delta-learning method (see, e.g., a nice review by Huang et al. <https://www.science.org/doi/10.1126/science.abn3445>), in which first computationally less demanding baseline NNP model is constructed (here using metaGGA XC DFT functional SCAN with D3 dispersion correction) on a full database (herein ~250k metaGGA datapoints) and then a correction (delta) model is trained on much smaller dataset subsampled from a full dataset evaluated at computationally more demanding level (in our case calculations using hybrid-level XC DFT functional wB97X-D3 and only on 150 data points taken from CHA proton jump simulations). *This is computationally much more efficient compared to training an NNP with wB97X-D3(BJ) as a reference directly ("from scratch") for a process/reaction of interest.* The reason is that the correction (delta) surface is expected (and is shown by this and many other use-cases in literature) to be much smoother than the full potential energy surface.

This work could be published after the authors address previous comments and improve the documentation of the supporting materials.

We thank the reviewer for the comments and suggestions and for the recommendation to publish this work after addressing these issues. We believe we have now addressed the reviewers' comments, significantly enhanced the supporting information and made the data sufficiently (freely and publically) available.

Reviewer #3 (Remarks to the Author):

Referee report Grajciar et al.

A reactive neural network framework for water-loaded acidic zeolites

The manuscript is situated in a very interesting research field, namely the derivation of machine learning potentials for reactive events in zeolites in presence of water. A huge amount of work has been done by the authors and the results are promising. However in its current form, the manuscript combines a lot of aspects and the authors give a series of bold statements on the final goal of the research which is not fully substantiated by a proper validation of the neural network potential.

We thank the reviewer for a generally positive evaluation and appreciation of our work and suggestions for its improvement.

We acknowledge that it is basically impossible to prove (although very easy to disprove) the generality of our NNPs across such a broad configurational and chemical space. Therefore we have softened our statements throughout the manuscript (in Introduction and Conclusions sections in particular). Nevertheless, we would like to highlight that we did test the transferability (as one Reviewer mentions a "huge amount of work has been done") extensively considering multiple challenging generalization/out-of-domain (OOD) tests covering a range of Si/Al, pore shapes and sizes, water concentrations and multiple specific reactions. Also we would like to highlight that the NNPs were designed to cover the breadth of configurational space by using a robust SOAP metric. We have now added multiple further tests in this revision to the manuscript (and SI), each of which turned out to support the validity of our initial rather "broad" statements. These tests include: i) the free energy calculation with

“vanilla” metadynamics of proton jump in H-CHA model - comparing the performance of NNP to fully first principles biased simulation at the reference-level employing SCAN+D3 XC DFT functional, b) the equilibrium MD simulations at the SCAN+D3 for a novel “pinned hydroxonium” species observed in during NNP MD runs complemented by single point (SP) calculations at the NNP level, c) the static water adsorption energies from geometry optimizations on all distinct T-sites (T1-T12) in H-MFI zeolite.

Lastly, we share with the reviewer(s) our preliminary data, which exemplify encouraging transferability/generalizability of our NNPs for description of defective species these data are a part of ongoing work in which we investigate confined water in (defective) zeolite FAU using an extended version of the Δ NNP model presented herein. The Figure below shows the results for a series of generalization (OOD) test simulations using the improved Δ NNP model for defect-containing FAU (unseen during Δ NNP training) containing either extra-framework (EFAL), framework-associated (“fwal”) aluminum species or a silanol nest. It turns out that the Δ NNP model shows excellent accuracy with respect to its ω B97X-D3(BJ) reference level (about 1.5 meV/atom and 110 meV/Å error for energy and forces, respectively).

Figure: a) Relative energy error ΔE distributions of the Δ NNP model with respect to the ω B97X-D3(BJ) level for three defect-containing FAU test cases: (b) extra-framework Al, (c) framework associated Al, (d) silanol nest.

Furthermore a rather extensive comparison is performed with ReaxFF. In the opinion of this reviewer, the parts on ReaxFF should be drastically reduced or even be fully removed from the main manuscript. ReaxFF is not a good comparative basis for neural network potentials as the latter have the goal to obtain results at quantum mechanical accuracy. Within the zeolite community ReaxFF is not often used, as it depends on a lot of parameters and can not be regarded as chemical accurate. In summary the manuscript in its current form, is designed to show that a lot can be done, without real validation.

We acknowledge the concerns of the reviewer (and the reviewer 1) and have therefore significantly reduced the discussion of ReaxFF performance to a very brief discussion in the section “Generalization tests and exploration of configuration space” (including Figure 1c) and the rest of comparisons with ReaxFF is now moved to SI. However, we would like to point out that, unfortunately, ReaxFF is the only reactive force field that covers the BAS-zeolite-water systems, i.e., making it the (only) relevant competitor - this motivated us to compare against it originally. Also, the vast majority of the verification/validation was done against the DFT as a reference already in the original manuscript and now we have extended this validation (against DFT) by further tests. However, as we responded to the previous related comment by the reviewer - unfortunately, it is basically impossible to prove (although very easy to disprove) the generality of our NNPs across such a broad configurational and chemical space - we tried to perform extensive and unbiased generalization (OOD) tests initially, which we supplemented by further tests in the revision stage and we hope that this would satisfy the reviewer along with the softening of some of our statements.

Furthermore the authors claim that in the current manuscript, a toolset is built to describe in a more general way reactive events in zeolites in presence of water, but it is unclear how the authors see the usage of the methodology? Is the derived neural network potential transferable in the sense that it can directly be used by other researchers working in the field of zeolite catalysis or is it the intention that other researchers working in the field of zeolite chemistry, derive their own potential based on the methods presented in the paper? If the final goal is the first one, it would be necessary that the potential is made fully available to the research community, together with necessary scripts to use the potential.

We thank the reviewer for a helpful suggestion regarding the clarification of our intentions. Indeed, our main aim is to provide a general NNP to be used by the other researchers in the field for their own projects and problems - **for that reason we have now made the potential(s) fully publicly available for all researchers** together with all the other data which should make our work reproducible.

A secondary aim is to provide use-cases (delta-learning and automatic collective variable generation) on how such a general NNP can be extended/tweaked. However, it is true that we do not provide a specific toolkit (library) to carry out all these extensions.

We have therefore rephrased a few sentences throughout the manuscript to define our intentions more clearly and added links to stand-alone libraries enabling automatic collective variable generation explicitly to the “Data availability” statement.

There are some concerns with respect to the data availability (see comments later in this report). It is thus not fully clear which message the authors finally want to bring to the community. In the opinion of this reviewer, it would be better to bring a more focused message with data that are thoroughly validated.

We acknowledge this shared concern amongst the reviewers and therefore we have made both the potential(s) and the full dataset fully publicly available (under CC BY-NC-SA license) which together with other data provided should make this work reproducible and the potentials easy to deploy by the community for the problems of its liking (<https://doi.org/10.5281/zenodo.10361794>).

More detailed comments are given below :

Abstract

– The abbreviation H-AS is not widely used within the zeolite field.

The abbreviation has been removed from the abstract and replaced throughout the manuscript with the more established BAS abbreviation standing for Brønsted acidic site zeolites.

– The sentence “we have developed a general potential surface interpolator with consistent accuracy” should be revised. It would be better to state that a machine learning potential for the Potential Energy Surface has been derived instead of using the terminology interpolator. It is not clear what is meant with consistent accuracy. The terminology chemical accuracy is used in the field to refer to methods where the errors on the energies are less than 1kcal/mol, one could also state that an accuracy the same as the underlying training data for example obtained at DFT level is obtained, but consistent accuracy is not the right terminology in this context.

We have rephrased the sentence in the abstract following the suggestions of the reviewer.

– “Furthermore, we show that the baseline model.... collective variable”. This sentence is confusing and showcases the divergent focus of the article.

We have rephrased the abstract significantly to clarify the focus of the article.

– “This framework allows for operando simulations ...” This statement is much too bold and should be toned down. The manuscript does not provide all tools to simulate realistic catalysts at operando conditions. Furthermore the terminology quantitative accuracy is again confusing, see the earlier remark on the definition of accuracy.

We have rephrased the abstract significantly to address the points raised by the Reviewer and the related comments below.

Introduction

– Page 3 : Last sentence and Page 4 first sentence. Very bold statements are made, the authors claim to cover all aluminosilicates, at all possible water compositions, aluminum substitutions and reactive events, thus they claim to develop a potential which is able to describe almost the whole field of zeolite catalysis, with the exception of metal-loaded zeolites. The results presented in the current paper, indeed show that the authors can simulate a lot of things, but in the opinion of this reviewer, the work does not show that the whole field of Brønsted acidic zeolites can be simulated with the work proposed in this manuscript.

We would point the reviewer to our more specific answers above to the similar concerns raised by the reviewer and here we only briefly recapitulate that we have softened our statements in the Introduction and included additional verification tests.

We would however like to highlight two main points that support our claims of generality. Firstly, the NNPs were designed based on a SOAP metric with the particular aim to cover the breadth of aluminosilicate zeolites, Si/Al ratios and water contents, and this attempt has been aggressively tested via in and out of domain verification tests across that space, including H-FAU in range of Si/Al and H₂O concentrations, H-MFI slab with bulk water; GIS melting with water; silicic acid and aluminum hydroxide in bulk water. It is this generality that provides our work with both its novelty and its value to the community.

Secondly, via additional verification tests, including H-MFI water adsorption energies for all T sites; free energy profiles for proton jump in H-CHA; equilibrium MD for novel “pinned hydroxonium” in H-FAU; silica database validation with diversity of frameworks including silica polymorphs; and even some EFAL species, each of these tests was found to support our claims.

We do however acknowledge that it is basically impossible to prove that the whole field of Brønsted acidic zeolites is covered. Hence, we have softened our statements in this section of the Introduction as well as in the other parts of the manuscript.

Results

Database generation and training of the general H-AS NNPs

– The motivation for the structures to be taken up in the training set is not clear. A rationale should be given why certain zeolite models were taken into account in the training data. The authors used three existing zeolites namely SOD, CHA and MVY and 7 hypothetical zeolites. Was it the intention to cover the whole zeolite atlas? Why did they choose SOD, CHA and MVY as existing frameworks? Is there any rationale for the structures in terms of the secondary building blocks needed to be present to account for the whole set of materials? More explanation should also be given on the hypothetical zeolites, which secondary building blocks do they contain? This motivation and clarification should be taken up in the main manuscript.

Indeed, our aim is to model the whole range of existing and hypothetical (alumino)silicate zeolites using NNPs with close to DFT accuracy. Therefore, we selected the initial structures for dataset creation to get high structural diversity of silica zeolites in terms of atomic density and similarity of atomic environments (measured by SOAP-based metric - the usefulness of which for mapping the zeolite configurational space has been shown before <https://doi.org/10.1063/1.5119751>). We first chose 7 zeolites by Farthest Point Sampling together with the SOAP descriptor (SOAP-FPS) from a hypothetical zeolite database which contains more than 330k siliceous zeolites. This procedure allows us to generate a subset of structures with maximally distinct atomic environments in the zeolite structure. Then, we also added three existing zeolites with different framework densities (CHA, SOD and MVY: 14.9 – 20.5 Si nm⁻³) and a low-density, two-dimensional silica bilayer (BL) to further improve the coverage of the zeolite configuration space.

To clarify these points, we have added a more detailed description of the database creation procedure to the Results part of the main text and Supplementary information including a list of framework densities and database indices of the hypothetical zeolites to Supplementary Table 1. Additionally, we have added Supplementary Figure 1 depicting the structural diversity of the zeolites used for training and additional test simulations (Supplementary Table 4, see comment below) in comparison with the hypothetical zeolite database. Also, we have added an analysis of the ring topology of the zeolites contained in the database (both existing and the hypothetical) to Supplementary Table 1 - following a recent work by Crum et al. (<https://doi.org/10.1016/j.micromeso.2023.112466>). Based on this analysis, the zeolites in the database cover a broad range of zeolite ring topologies from 3-membered rings all the way to 14-membered rings (with only a very unique 13-membered rings missing). All initial structure models are also available in the zenodo repository (<https://doi.org/10.5281/zenodo.10361794>).

– The authors took also slab systems into account, it is unclear how these were constructed and terminated. How do these models compare with slab models that were modeled by Chizallet and co-workers or with models taken up in the database of Sholl (10.1021/acs.chemmater.8b03290).

The explanation of the slab generation procedure was already included in the original version of the manuscript. Nevertheless, we have added references for closely related surface models (MFI and silicatene) including a more detailed description have been added to the main text (Methods section) and SI.

– It seems that the simulations to generate the training data set were performed at very high temperatures from 1200K to 3600K, which is much higher than any catalytic process. Some catalytic processes take place at high temperatures e.g. 700-800 K. It is motivated that high temperatures are necessary to sample reactive events, but in how far are the materials still stable at these conditions?

The temperature ranges considered in the generation of training data indeed extend beyond catalytic application temperatures (FCC process ~1000 K) and beyond silicate glass melting temperatures (onset T ~2000 K). However, this is not a problem. Indeed, the instability of the

materials (i.e. via the breaking of covalent bonds) is the aim of the high temperature simulations.

Generating reference data using high temperature (AIMD) simulations is a common practice in the machine learning potential (MLP) field (see, e.g., Erhard et al. <https://www.nature.com/articles/s41524-022-00768-w>, Bocus et al. <https://doi.org/10.1038/s41467-023-36666-y> or Csanyi et al. <https://aip.scitation.org/doi/10.1063/5.0013826>) to improve sampling of the configurational space, including sampling of variety of highly activated reactive events such as (covalent) bond breaking and formation. MLPs trained on such reference data sets are in general more robust/transferable despite exhibiting slightly decreased accuracy. Thus, by deliberately increasing the probability of sampling such events in the training dataset, we improve the generality of the NNPs and their ability to model such reactive processes.

Nevertheless, we have rephrased a few sentences in the Results section to clarify that this is a common practice in MLP development adding relevant references.

– Varying Si/Al ratios are considered. Also here a motivation should be given of where Aluminum was placed and why, the number of symmetrically equivalent Aluminum sites, the proximity of Aluminum etc. Aluminum siting is a very challenging topic in the field of zeolite catalysis, it would be perfectly understandable if the authors would not be able to cover the whole range of Si/Al ratios, but in the current form of the manuscript it is claimed that they cover the whole space of Si/Al ratios, therefore it is necessary to address these issues, otherwise they need to tone down the statements made in the manuscript.

We thank the reviewer for the relevant suggestion. We added a more detailed explanation of the procedure of generating structures (including those with different Si/Al ratios) and their subsampling to the main text and the SI (Methods section in the main text and the Supplementary methods section in the SI). In addition, we considered a new generalization (OOD) test, in which we calculated the water adsorption energies on all distinct T-sites (T1-T12) in H-MFI zeolite (see Supplementary Figure 7 and the related discussion in SI). For each of these structures a separate geometry optimization was done at NNP, SCAN+D3 and PBE+D3 levels. The NNP is stable and provides reasonable water adsorption energies (between 80-110 kJ/mol) with the mean absolute deviation (MAD) between NNP and SCAN+D3 is about 10 kJ/mol, and the NNP is in most cases able to follow the relative ordering of stabilities observed at the SCAN+D3 level. In this context, we would like to highlight that MAD for PBE+D3 wrt. SCAN+D3 is about 12 kJ/mol, i.e. the NNPs are at least as good as other “flavors” of the XC DFT functional.

Regarding the process of generating the distinct aluminum structures, initially, for all the frameworks considered in the database (chosen by the furthest point sampling - FPS - algorithm using SOAP-based metric) the aluminum distributions were generated **randomly** (just following the Loewenstein rule) covering low (typically one Al per unit cell), medium and high Al (Si/Al<2-3) concentration - the idea was not to bias it towards a particular T-site type. Running high temperature simulations together while considering rather harsh deformations from equilibrium positions (see Supplementary methods), we expect to cover local T-site environments that should deviate significantly from the equilibrium structural characteristics of a specific T-site (e.g., Si-O(H)-Al/Si angles). In other words, we expect to describe a broader range of T-sites (and frameworks) than those which we explicitly considered explicitly in the database. After the high-T AIMD and deformations, we carry out FPS with the SOAP structure-descriptors *focusing explicitly on the Al-environments* with the structures selected by the dissimilarity of the “Al-centered” environments -> this allows us to sample only the most diverse configurations of aluminum. Also we note that we also tested and validated the ability of the NNPs to cover different Si/Al ratios (and frameworks) by running simulations for OOD models of FAU (FAU(Si/Al=1)+48H₂O and FAU(Si/Al=47)+1H₂O) and GIS (Si/Al=1+24H₂O).

– Is it correct that no explicit enhanced sampling simulations were done to generate information for the training of the neural network potential? As the authors claim to cover the

whole space of reactive events, were enough reactive events taken up by performing the superhigh temperature simulations? In this case the authors should comment more on the reactive events they observed during the high temperature simulations. In the current version of the manuscript and supporting information nothing is mentioned about these aspects.

As mentioned in the Methods section, we have used a small amount of enhanced sampling simulations in our dataset, namely biased AIMD runs of Si-O(H) and Al-O(H) bond cleavage mechanisms in H-CHA at high-water loading conditions taken from reference Heard et al. (<https://www.nature.com/articles/s41467-019-12752-y>, <https://doi.org/10.1039/C9SC00725C>) - we added this now explicitly to the "Database generation and training of the general BAS zeolite NNPs" section.

In the high-temperature AIMD simulations in the training dataset, similarly to the the generalization test for GIS (Si/Al=1+24H₂O) discussed in the section "NNP performance for highly activated reactive event", we have observed thousands of reactive events (e.g., Si-O and Al-O bond hydrolysis, aluminol and silanol formation, increase of Al coordination up to six, water splitting/re-formation, water autoprotolysis, (de)protonation of silanols/aluminols, formation of aluminum oxide-like island in the framework, etc.). We stress here that our aim was not to select for specific reactions manually but to let the high-temperature simulations sample/explore the relevant events in an unbiased way (see also our comments on using high-temperature simulations above). We have described in more detail the type of reactive events observed in these simulations in this section and in the "NNP performance for highly activated reactive events" section. We would also like to point the reviewer to the data, which had been already originally included in the manuscript - the Supplementary Figure 5 (originally SI Fig. 4) and the discussion of the MFI test case results for more details on the type of reactive events we observed.

Lastly, we would like to make a short comment on the philosophy behind this approach - it is a power of the machine learning that one does not need to manually specify the structures/reactions of interest and track all the transformations visually - the aim is to allow the (high temperature) simulations to sample broad chemical and configurational space without an unreasonable bias (temperature being a very unspecific bias exciting all the degrees of freedom) and then use a robust configuration space metrics to subsample the explored space automatically (without a need to visually check the structures and select), thus limiting the data redundancy and unnecessary human bias.

– How much training data were eventually used for the generation of the neural network potential?

The dataset size (248439 structures equipped with forces and energies) was already mentioned in the Methods section. We have now also included it in the corresponding paragraph of the Results section in the revised manuscript.

– Why did they authors use SchNet architecture? Much more efficient mathematical frameworks are available, which allow to reduce the number of training data substantially and also allow to reduce the errors.

We are aware of the new more data-efficient architectures of neural networks (in particular the rotationally equivariant NNPs such as those introduced by Batzner et al. - <https://www.nature.com/articles/s41467-022-29939-5> or Schutt et al. <https://arxiv.org/abs/2102.03150>). However, for historical reasons, i.e., large quantities of production simulations carried out already with the SchNet architecture including multiple time-consuming testing and validation, we opted out to keep using the SchNet architecture throughout this work. Also, the aim of this contribution is mostly to show that the development of such transferable NNPs is possible. But of course, using rotationally equivariant NNPs one can improve data-efficiency and accuracy. Therefore, we are already testing these new architectures and are considering them in our future work.

– The t-SNE plot is instructive, however the authors should comment in how far they are able to describe diverse topologies of the zeolite database and currently in the figure only FAU, MFI, etc is indicated. Is the whole zeolite database well described?

To test the NNP accuracy for modeling the structural diversity of zeolites, we have added Supplementary Figure 1 and Supplementary Table 4 with (all-silica) test cases to evaluate the NNP accuracy with respect to DFT and experiment, also in comparison with our previous NNPs for siliceous zeolites. These zeolites are a random sample of 1000 topologies, drawn across the breadth of energy/density space. The NNPs show good agreement with both DFT and experiment with similar accuracy as our previous all-silica NNPs (see also responses to Reviewer 1). Thus, we conclude that indeed the NNP is able to describe diverse topologies well. We have added a discussion of these results to the Results section (“Sampling of equilibrium properties”) of the main text and to the SI.

– In terms of validation, the authors refer to Table 2 in the Supplementary information. More information should be provided in the construction of the training and validation set.

We have added additional details of the training and validation data to the Results part (“Generalization tests and exploration of configuration space”), the Methods section of the main text and to the Supplementary Information (see also response to comments above and below).

Generalization tests and exploration of configuration space

– It is suggested to remove the comparison with ReaxFF completely from the main manuscript and focus on the performance of the NN potential with respect to the underlying DFT data. It is obvious that that neural network potential is much better than the ReaxFF, this can be mentioned once and if necessary the results on ReaxFF can be taken up in the SI.

We acknowledge the concerns of the reviewer and have therefore significantly reduced the discussion of ReaxFF performance to a very brief discussion in the section “Generalization tests and exploration of configuration space” (including Figure 1c) and the rest of comparisons with ReaxFF is now moved to SI (see responses above for more details).

– Table 1 : How were the validation systems generated? For example FAU+48 H₂O molecules, how were the structures of this systems determined? From MD simulations? If yes, how long were the simulations? A lot of details are missing for the validation protocol.

Details of the structure generation and MD simulations were already part of the Method section. We have now also added additional information on the initial structure generation and MD simulation setup to the Method section and added links to the Methods section in the corresponding paragraphs of the Results section.

– Table 1 : it is unclear what the reference was for the errors. Are the methods validated with respect to underlying DFT data?

All validations mentioned in Table 1 are with respect to the reference DFT calculations at the SCAN+D3(BJ) level. We have amended the table caption to clarify this.

– It is suggested to completely remove the rather extensive discussion on ReaxFF in the main manuscript.

We have removed the ReaxFF discussion from this section (see responses above for more details).

Sampling of equilibrium properties

– Reference is made to [46] a paper on chemrxiv (<https://doi.org/10.26434/chemrxiv-2022-d1sj9>), which does not seem to exist.

The hyperlink of reference mentioned by the reviewer has been corrected and is now: <https://doi.org/10.26434/chemrxiv-2022-d1sj9-v3>.

– Page 10 : simulations on FAU with various Si/Al ratio and water loadings. From the discussion it is unclear which simulations are performed with the NNP and DFT? Are DFT simulations done for validation? For example an adsorption energy of -79 kJ/mol is reported, probably this value was obtained with the NNP? Was the DFT adsorption energy also calculated? The authors refer to reference 45, which suggests that the DFT simulation was not performed? The same remarks hold for the further discussion, where the degree of solvation of the BAS is discussed. Qualitative features are discussed on the degree of solvation obtained with the NNP, but it is not clear in how far these observations correspond to simulations performed at the DFT level. It only shows that the NNP is stable during the simulations.

In this part of the “Sampling of equilibrium properties” section, we have not done dynamical DFT simulations to calculate water adsorption energies - we refer to DFT values reported in the literature - both former reference 45 (now 52) and now we have added also a reference to work of Grifoni et al. (<https://www.nature.com/articles/s41467-021-22936-0>). We have rephrased a few sentences in this section to clarify this point. But we note, that we did carry out DFT (SCAN+D3) single point calculations using data from these NNP dynamical runs and validated the accuracy of the NNPs with respect to SCAN+D3 in this manner (see Table 1). We would also like to stress that both the water adsorption energies and qualitative features (solvation degree) of water dynamics reported in the literature using DFT simulations (geometry optimizations or (biased) MD) are in *qualitative agreement* with the NNP results presented herein and the minor deviations are expected to be well within the range expected from the use of different XC DFT functionals in the literature (and which does not, in our opinion, justify a need to run additional costly dynamical calculations at the reference metaGGA SCAN+D3 to obtain converged values of water adsorption energies). This shows, in our opinion, much more than just that the NNPs are stable during simulations. Again, we have rephrased a few sentences in this section to clarify this point.

Lastly, this comment from reviewer 3 (as well as related comments from reviewer 4) motivated us to carry out dynamic simulations at the DFT level for a “pinned hydroxonium” species (see Supplementary Figure 6 and related discussion), observed initially during our NNP calculations for (FAU(Si/Al=1)+48H₂O) system. The “pinned hydroxonium” species remained stable during 10ps duration SCAN+D3 AIMD, the NNP errors both in energies and forces remain small (similar to Table 1 reports for FAU(Si/Al=1)+48H₂O simulated dynamically at the NNP level and subsampled with SCAN+D3). Furthermore, the covariance analysis (see SI Figure 6 and SI Table 3) supports the notion that the NNP indeed reconstructs broad parts of the DFT-based PES well and is thus able to sample similar configurational/phase space as well as the DFT reference. We added a brief discussion of the “pinned hydroxonium” and its consequences to this section of the manuscript with the details included in the SI.

– Page 11 : “In both cases, the NNPs accurately reproduce the SCAN+D3(BJ) energies and forces.” This sentence seems to suggest that DFT validation runs were performed. Can the authors clarify which simulations have been done in production with the NNP and which validation runs have been performed?

In this particular case of validation MD runs, the NNP forces were used to propagate the systems in time and DFT single-point (SP) calculations were carried out for configurations uniformly sampled from the MD trajectory and NNP and DFT energies and forces for these snapshots were compared and the errors evaluated and depicted in Table 1.

We have added more details about the NNP validation procedure to the Results part in the initial paragraph of the “Generalization tests and exploration of configuration space” section (see also response to comments above).

NNP performance for highly activated reactive events

– Simulations are performed at extreme conditions, namely 1600K and 3000 K, to force reactive events. Such simulations are not very realistic. The authors do observe some reactive events like the formation of silanol defects etc. These simulations prove that the system remains stable under these extreme conditions but there is no validation that the NNP yields realistic chemical processes. On the positive side, the authors do validate the energies and forces for a series of snapshots during these simulations and observe that the errors remain within acceptable range. However to truly validate the NNP it would be necessary to simulate some free energy barriers for

some chemical processes both with the NNP and the underlying DFT level of theory. It would be necessary to validate that the DFT methodology and the NNP samples similar parts of phase space and obtains similar free energy barriers for the activated processes.

We acknowledge that high-temperature conditions *per se* are not very realistic. However, this does not mean that the reactive events observed during such simulations are not realistic - the high temperature mostly increases the probability of such rare events to take place and/or potentially just re-orders the relative rates of these otherwise realistic processes with respect to each other. On the contrary, e.g., the reaction events observed for the “siliceous MFI slab with bulk water” model appear to be reasonable and mostly belong either to common knowledge ((water autoprotolysis) or have been observed previously in the literature (e.g., Heard et al. - original ref 15, Jin et al. - original ref. 54) such as proton shuttle via Grothuss mechanism, OH- attack on silicon creating penta-coordinated Si-species, or rupture of Si-O-Si bond in the penta-coordinated Si-species via deposition of proton. We have amended a few sentences in this section to clarify this.

Nevertheless, we acknowledge the usefulness of showing explicitly that the NNP is able to simulate free energy profiles of reaction processes in agreement with the underlying DFT level of theory. Therefore, we have simulated *a proton jump process in the H-CHA model* (between O2-O3 oxygens) using a “vanilla” *metadynamics simulations* at both the very costly reference metaGGA SCAN-D3 level (above 60 ps with a timestep of 0.5 fs) and the NNP level. We show that the NNP is able to reproduce the reference DFT free energy profile well, with only minor deviations (errors within 10 kJ/mol). Together with the other newly added validation tests (“pinned hydroxonium” dynamics and covariance analysis) it clearly suggests that the herein-developed NNP indeed reconstructs broad parts of the DFT-based PES well and is thus able to sample similar configurational/phase space as the DFT reference. We have added a discussion of these new results on the metadynamics simulations to the “NNP performance for highly activated reactive events” section including reformatting of the Figure 3, with additional technical details provided in the Methods section and in the Supplementary Information.

Lastly, we would like to add that besides the standard validation process (see section “Generalization tests and reaction path searches” and Table 1) we also tested NNP performance on *particular reactive events/mechanisms* (water-assisted bond breaking mechanisms of the Si–O and Al–O(H) bonds) using NEB calculations.

Hence, all-in-all we carried out a battery of various validation/generalization tests which all support our claims that NNPs are able to sample also the activated reactive events.

Extensions of the NNP model

– The delta learning procedure is promising, however more details should be given on the methodology.

We thank the reviewer for a positive evaluation of the concept. A more detailed description of the delta-learning approach has been added to the beginning of the “Improving the baseline

model accuracy using delta-learning” section. Also, this whole section has been thoroughly rewritten and improved (see also comments below).

– Figure 3 : It seems that the errors made by the delta learning procedure are of the same order of magnitude than the delta between the levels of theory. This needs to be clarified and it needs to be validated in how far the methodology would work for levels of theory which are further apart.

We acknowledge that the original version of the subsection “Improving the baseline model using delta-learning” did not successfully highlight the added value of this approach (in conjunction with the robust baseline NNP model) and could lead to confusion about its usefulness. Therefore it has been now thoroughly revised following also the comments from the other reviewers.

In particular, we highlighted much more clearly the excellent performance of the delta-learning model for the *in-domain* case, i.e., a proton jump in H-CHA on which the Δ NNP model was trained, which is a common application domain/target of the Δ NNP model (and which we did not stress in the original version).

Then, we focus on much more challenging (and interesting) cases, in which we test the ability of the Δ -NNP model to generalize to out-of-domain cases such as proton jump in different framework (H-FAU) and a completely different reaction in different framework (Al-O(H) bond dissociation in H-FAU). For such cases it is not surprising that the errors increase, but it is surprising and important that the use of the Δ NNP model is still beneficial for proton jumps in a different framework (H-FAU) or that it does not worsen the description (over baseline model) for a different reaction in a different framework.

It is true, as the reviewer points out, that in the specific case of framework hydrolysis in H-FAU, the errors of the delta-model are of the same degree as those between the levels of the model. However, this is in part an artifact of the similar predictions made between the two functionals. The important point is that the delta model remains stable so far out of domain, rather than that it provides a reduction in error for this particular case.

Lastly, it might be relevant to point to some of our encouraging preliminary results mentioned above in the context of the transferability/generalizability of our NNPs for description of few examples of the defective species (see the answer to the first comment of the reviewer 2), which also showcase the robustness/potential of the Δ -NNP model for much more challenging/general use cases.

– The part on the acceleration of the rare events is interesting but confirms the general statement that a bit of everything is collected in this paper without real focus. It is also questioned in how far the procedures are made available for the rest of community. If it is the intention to show that a lot can be done by the methods described in the paper, it would be necessary to provide the necessary tools for the rest of the community to actually use the methods.

We have answered above a similar comment by the reviewer - we have rephrased a few sentences throughout the manuscript to define our main intention clearly, which is to provide and verify the use of a general NNP, to be used by the other researchers in the field for their own projects and problems.

Regarding providing the tools to the community, we have now made the potential(s) fully publicly available for the academic community.

In addition, the python package to generate collective variables (<https://github.com/martinsipka/aucol>) along with modified plumed library (<https://github.com/martinsipka/plumed2>) has been now directly mentioned in the “Data Availability” statement.

Conclusions

– I reiterate some of the statements formulated above, very bold statements are made, namely that the tools will enable to simulate hydrothermal instabilities, formation of defects, etc under operating conditions. It is not clear which tools are made available for the community and in how far these are sufficiently validated to be used in a more general context.

We thank the reviewer for a thorough review and for many important suggestions/comments that have improved our contribution. We believe that, in our responses above, we have addressed the issues raised by the reviewer sufficiently (with many new calculations/tests added to the manuscript/SI and to this response as unpublished material) and described the changes made in the manuscript and the SI as a consequence of the issues raised.

Data availability

It is unclear which data are made available. It seems that the potential is not provided in the zenodo database? Following statement is mentioned on the zenodo link : Trained Neural Network Potentials NNPs including the Δ -learned model (SchNetPack, version 1.0) are available under the CC BY-NC-SA 4.0 license for non-commercial use only. Furthermore following statement is given You may request access to the files in this upload, provided that you fulfil the conditions below. The decision whether to grant/deny access is solely under the responsibility of the record owner. It is thus unclear whether the potential is made available or not? Related to this point, reference is made to the earlier remark in this report, namely what is the final intention of the manuscript? Providing a potential that can be used by the research community or showcasing a methodology to be used by other researchers?

Irrespective of the final goal, it is necessary that all information including the NNP, the training data, the validation set and scripts are made available for the research community.

We acknowledge this shared concern amongst the reviewers and reiterate that our goal is to test and provide a tool for use among the scientific community. The aim in restricting access upon request was based on an attempt to avoid unauthorized commercial use. This has now been resolved as we have made both the potential(s) and the full dataset fully publicly available (under CC BY-NC-SA license) which together with other data provided should make this work reproducible and the potentials easy to deploy by the community for the problems of its liking (<https://doi.org/10.5281/zenodo.10361794>). We note here that we provide considerably more data than are required by the standards of the field or journal, as we aim to achieve maximum utility for the community.

Reviewer #4 (Remarks to the Author):

In the submitted manuscript, a reactive neural network potential (NNP) reproducing the reference metaGGA DFT level is proposed for describing zeolite systems, possibly loaded with water. The utility of the potential is demonstrated on a number of examples that involve the sampling of equilibrium states (water adsorption in faujasite and chabazite) and description of activated processes (surface defects formation) in static and MD calculations. Furthermore, the use of Δ -machine learning for accuracy improvement (via changing the electronic structure method) is proposed and demonstrated on static and metadynamics calculations of reactions of proton transfer and Al-OH bond dissociation in faujasite. The manuscript is well written, the scientific presentation is sound, and the practical examples presented provide a clear demonstration of the usefulness of the approach. I recommend its publication in Nature Communications provided the comments listed below are carefully addressed.

We thank the reviewer for a positive evaluation of our work and very useful suggestions for its improvement, which we address below, accompanied by a brief description of the changes made in the manuscript and the SI as a consequence of the issues raised.

1.) According to data presented in the paper and SI, the mean absolute error achieved for forces is of order of 0.1 eV/Å. I wonder if this is sufficient accuracy for the applications discussed in the manuscript (relaxations and MD simulations). In relaxations, for instance, the relaxation criterion for forces used in practice is of order of 0.01 eV/Å or less and, clearly, a noise that is an order of magnitude greater than this is likely to confuse the relaxation algorithm or even to yield structures that are significantly different from the structures relaxed without the use of machine learning potentials. The authors should clearly explain what their requirements for accuracy of the NNP are and what motivated their choice.

We thank the reviewer for a very interesting point raised here (as well as the closely related comment below) and we address this point in multiple places in the manuscript now via additional simulations that compare DFT and NNP head-to-head for geometry optimization and (biased) MD simulations.

Our “phenomenological” answer is the following: our experience is (and since such force errors are commonly reported throughout the MLP field we expect it to be of more general validity) that even NNPs with such force errors (with respect to the reference) that are rather high when compared to standard relaxation criteria, are able to provide structures and properties (using either geometry optimizations or MD simulations) that are typically at least in the qualitative agreement with the reference.

We have not provided in the original manuscript specific examples of “head-to-head” comparison between NNP- and DFT-driven/optimized structures/properties as we have rather focused on comparisons via DFT single point calculations. Therefore (and also based on the comments of other authors), we have added multiple tests that aim to provide such “head-to-head” comparisons, namely: i) reoptimization of the subset of purely silica polymorphs and zeolites with experimentally observed properties (energies, densities, average Si-O bond distances, lattice parameters) - see Supplementary Table 4; ii) evaluation of the static water adsorption energies on all distinct T-sites (T1-T12) in H-MFI zeolite (see Supplementary Figure 7), in which separate geometry optimizations were done at NNP, SCAN+D3 and PBE+D3 levels; iii) the free energy profile calculation using “vanilla” metadynamics of proton jump in H-CHA model comparing performance of NNP to fully first principles biased simulation (see Figure 3e and SI Figure 12) at the reference-level (SCAN+D3). Also we considered the equilibrium MD simulations at the SCAN+D3 for a novel “pinned hydroxonium” species observed in during NNP MD runs, which we complemented by SP calculations at the NNP level (see SI Figure 6). In general, these tests, along with some other tests we performed previously for the same type of NNPs such as VDOS calculations (Erlebach et al, <https://www.nature.com/articles/s41524-022-00865-w>), show that the structures/properties (adsorption heats, free energy profiles, lattice parameters, densities, average Si-O bond distances, etc.) obtained by NNP are at least in qualitative agreement with the reference-level (SCAN+D3) simulation. Or in general, considering the differences between the performance of different “flavors” of the XC DFT functionals (e.g., PBE vs. SCAN), the NNPs tend to perform as well (or as poorly) as different XC DFT functionals.

We expect that at least some of the issues observed (such as occasional suboptimal ordering of rotamers, noisy forces leading sometimes to incorrect local minima) can be resolved partially by adoption of rotationally equivariant NNPs (possibly extended by NNPs with long-range dispersion correction explicitly included), to which we are currently switching. These architectures have been shown to provide smaller force errors and are able to take as the input not only distances (scalars) but also distance vectors (and other higher order tensors if needed).

Regarding the possible reason for why such relatively high force errors are still “good-enough” we point to the reviewer’s question and suggestion for analysis below, which allows us to hypothesize about this point.

2.) Related to the previous point - it is not clear from the presented data (e.g., Supp. Tab. 2) to which extent the error in forces/energies correlates with the forces/energies from explicit DFT calculations. This is very important, because, if the error in forces is not random (i.e.,

uncorrelated to the actual force), the NNP will bias the sampling of configuration space. To this end, a careful covariance analysis would be useful.

We thank the reviewer for a very useful point and suggestion. We carried out covariance analysis for all generalization tests (see Supplementary Table 3) and also for a new “pinned hydroxonium” model (see Supplementary Figure 6), in which SCAN+D3 AIMD is complemented by SP calculations at the NNP level. The average covariance (averaged over all generalizations tests) between the actual forces (at DFT) and the NNP error in forces is small (0.13), which indicates that errors in forces are mostly random (the largest force covariance was obtained for FAU(Si/Al=1)+48H₂O system reaching 0.24). The energy covariance is somehow larger (average of 0.31) so one may expect a mildly increased energy error for high energy structures.

Merging now the discussion and results related to the current and the previous point, one may hypothesize that potential energy surface (PES) of the NNP appears to be a more noisy (or possibly smoother) copy of the actual reference (DFT) PES able to rather well describe the location of the local extrema (minima, TS, ...) as well its (average) curvature up to rather high energies/forces (e.g., allowing stable and rather accurate reactive MD trajectories even at very high temperatures - and thus high thermal energy fluctuations - such as in 2D-MFI case). As a simplification, one may visualize NNP and the reference PES as a pair of double-well potentials that are identical with the exception that: i) a rather random noise is added to one of them, and that ii) accuracy of NNP double-well mildly decreases as energy increases (but with the average curvature being mostly retained). This might be the reason for low covariance (in forces) and why such relatively high force errors are still good enough (see discussion above).

We have added a short discussion of the covariance analysis and its consequences to the manuscript along with the links to the data provided in the SI.

3.) The data presented in Tab. 3 compare the results obtained using NNP and Δ NNP with the explicit DFT calculations for the Al-OH bond dissociation in FAU. Given the data are meant to demonstrate how the NNP can be used for “Improving the baseline model”, the numerical results are, in my opinion, rather unconvincing, because the error in the relative positioning of stationary states on PES are as large as the differences between the two electronic structure methods discussed. In other words, the SCAN+D3-based NNP results seem to be as good/bad proxy to the explicit ω B97X-D3 calculations as the ω B97X-D3-based Δ NNP making the Δ NNP useless. For the state “TS1”, for instance, the explicit ω B97X-D3 yields 85 kJ/mol, while the ω B97X-D3-based Δ NNP and the SCAN+D3-based NNP models predict 93 and 77 kJ/mol respectively, i.e., the error with respect to the explicit result is the same for both models. Even worse, the positioning of the state label “I” obtained with the explicit ω B97X-D3 is 60 kJ/mol, the ω B97X-D3-based Δ NNP model overestimates this value by 8 kJ/mol, while value obtained from SCAN+D3-based NNP model is only 3 kJ/mol lower than the explicit ω B97X-D3 result.

The authors should explain better why they think their Δ NNP procedure is useful.

As we noted in the answers to similar point by the other reviewers, we acknowledge that the original version of the subsection “Improving the baseline model using delta-learning” was not highlighting the added value of this approach (in conjunction with the robust baseline NNP model) and could lead to confusion about its usefulness. Therefore it has been now thoroughly revised following also the comments from the other reviewers.

In short, a standard application domain/target of the Δ -NNP models are the *in-domain cases*, e.g., such as a proton jump in H-CHA, on which the Δ -NNP model was actually trained. The idea is to train Δ -NNP model on a very small dataset for a *specific* process/mechanism of interest (here proton jump in H-CHA) and then apply it for this *specific* process to accelerate sampling. This is what we did not stress in the original version of the manuscript. This is now resolved, and is sufficient to explain why the delta procedure is useful.

Instead we originally focused on much more challenging cases, in which we wanted to test the ability of the Δ -NNP model to generalize to out-of-domain cases such as a proton jump in different framework (H-FAU) and a very different reaction in different framework (Al-O(H) bond dissociation in H-FAU). By focusing on out-of-domain cases, as correctly noted by the reviewer, we created confusion about the usefulness of the procedure.

Also, we would point out that the Al-O(H) bond dissociation model is a challenging case also from another perspective - the differences between SCAN+D3 and ω B97X-D3 predictions are rather small (within 7 kJ/mol) and thus very likely within the baseline NNP (and thus also Δ -NNP) accuracy of about 5-10 kJ/mol (see, e.g., Tables 2-3 or new metadynamics data for proton jump in H-CHA).

Lastly, it might be relevant in this context to point to some of our encouraging preliminary results about the transferability/generalizability of our NNPs for description of few examples of the defective species (silanol nests, extra-framework aluminum species and framework-associated aluminum species) that we discuss above in our answer to reviewer 3 and which were evaluated specifically for the Δ -NNP model.

4.) Page 27: "To avoid simulating degrees of freedom that are unimportant for the Al-O(H) bond dissociation reaction we introduced some restraints to the biased dynamics. We required the distance between the free water molecule and the aluminium atom to be at most 2.2 Å to avoid it diffusing away. We also fixed two hydrogen atoms to their corresponding oxygens to avoid permutations that would complicate the process. In the same fashion we disallowed the formation of the hydrogen bond (Figure 3d-FS) for different O and H pairs." I disagree that the restrained degrees of freedom are unimportant for the Al-OH dissociation reaction. Clearly, restricting the distance between water and Al significantly affects the entropy contribution to free energy of the given state. In other words, "diffusing away" is a way how entropy stabilizes the system and, when avoided, the sampling is biased towards states that might be irrelevant at the given thermodynamic conditions.

We thank the reviewer for careful reading. Indeed, we agree with the reviewer and we have now rephrased this part to make it clear that the constraining is an approximation that we have used to improve the sampling the reaction mechanism and which is a source of inaccuracy in the reported free energy profile(s).

5.) The data presented in Supp. Tab. 6 are not quite clear to me – what is the meaning of MAE and RMSE for reaction energy (E_r), which is a single value obtained either as potential energy difference between two minima (relaxations) or difference between ensemble averaged potential (or total) energies of two stable states? For such a quantity, NNP should yield a single number, and hence, trivially, the MAE and RMSE should be identical (equal to absolute value of difference between the E_r from explicit and NNP calculations). Likewise, the way the F and ΔE were obtained should be presented in more detail – the present version of SI is very scarce about that (just one sentence).

We used two energy error metrics in this work to evaluate the accuracy of the (Δ)NNP models with respect to their DFT reference. First, a reaction energy error ΔE_r (of a formation reaction in Eq 1 of the manuscript) to compare test systems across the entire chemical BAS-zeolite space. And second, the error $\Delta \Delta E$ of the relative energies ΔE with respect to a reference structure with the same chemical composition, e.g., the initial structure of an MD trajectory or NEB calculation. The use of ΔE_r allows evaluation of the NNP quality for transformations involving a change of chemical composition (adsorption, reaction or cohesive energies) while $\Delta \Delta E$ only quantifies NNP errors for the specific chemical composition or better a specific thermodynamic state point (see, e.g., <https://www.pnas.org/doi/abs/10.1073/pnas.2110077118>) only. Therefore, ΔE_r is larger in most cases due to a small ΔE_r offset of the NNPs caused by a (small) energy error of one of the reference structures (quartz, alumina, gas-phase water).

In the former Supplementary Table 6 (now SI Table 7), both energy and force errors are evaluated for 200 structures subsampled from the steered dynamics runs for the the Al-O(H) bond breakage and proton jump mechanism in FAU shown in Figure 3h-i (OOD test cases for the Δ NNP model) for which the SP calculations at the ω B97X-D3(BJ) are performed, i.e., energy and force errors are evaluated on a set of structures, for which the use of statistical measures (such as MAE or RMSE) is relevant.

We have added a more detailed description of the error metrics to the “Generalization tests and reaction path searches” section and into the SI including a more in-depth discussion of Supplementary Table 7 (former Supplementary Table 6).

6.) Some of the items in Supp. Tab. 7 are not unitless but their units are not provided

Units have been added to Supplementary Table 8 (former Supplementary Table 6).

REVIEWER COMMENTS

Reviewer #1 (Remarks to the Author):

I think the authors have done a good job in responding to the questions I raised, and most importantly they now share the training data and models produced, which is in agreement with research data policies.

I think this is a good manuscript, and deserves to be published. It's (in my view) too technical and not novel enough for Nature Communications, however (for the reasons highlighted in my original report and those of the other reviewers).

Reviewer #2 (Remarks to the Author):

The authors have improved the manuscript. This could be published as is.

Reviewer #3 (Remarks to the Author):

The authors made major efforts to improve the manuscript, which is appreciated. However there remain some concerns as detailed below.

Whereas some statements have been rephrased in the abstract, some sentences are still very general such as "we have developed a general reactive ... including the full range of experimentally ..". This remains a very bold statement, certainly when put in perspective to some other general purpose machine learning potentials which have recently appeared such as the MACE-MP-0 model, which furthermore seem to have lower errors for the trained energies and forces. The MACE-MO-0 model has MAE of 20meV/atom and 45 meV/Angstrom for the energies and forces (see also some remarks later in this report on the errors). The currently proposed neural network potential certainly has its added value, but it must be situated in a range of similar efforts to train general purpose machine learning potentials. I suggest that the authors add a paragraph to situate the work better in the overall efforts currently taking place in the field.

It is also suggested to change the sentence : "This NNP combines dramatic sampling acceleration" to "This NNP has the potential to dramatically improve sampling." In the opinion of this reviewer, the sampling can be accelerated as the cost to evaluate the forces is dramatically reduced with a machine learning force field.

Page 3 : The sentence " A state-of-the-art tool for accelerating the reactive sampling in"... should be changed. ReaxFF can certainly not be regarded as a state of the art tool for simulating reactive events in zeolite chemistry. This point was also made in the earlier review round.

Regarding the temperatures used for training : it is certainly true that using higher temperatures for training is commonly done, however the temperatures of 3600 K seem to be excessively high also compared to other literature references. As it is written now, it is suggested that simple increase of temperatures is a good methodology to sample more efficiently phase space for activated events, whereas more advanced enhanced sampling methods are generally necessary to efficiently explore phase space. The authors also address partly these issues on page 30, however the authors should comment on this point of sampling so ensure that reader is not misled.

There are some concerns regarding the errors of the energies and forces. The authors mention errors of ca 5 meV/atom and 186 meV/angstrom for the energies and forces respectively, which seem to be substantially larger than errors acceptable for machine learning potentials, currently being devised. On page 6, it is mentioned that these errors are similar to other reactive rotationally invariant MLPs and about the same order of magnitude compared to their own work. With current generation MLPs, errors are normally lower. The authors should comment on this, if the obtained errors are of this scale due to the usage of rotationally invariant network, they should

clearly state this. In general terms, it is advised that they add a paragraph mentioning what the acceptable errors are with the current generation MLPs. With the given errors, the NNP will probably be good to obtain a first insight on possible reactive events within zeolite catalysis, but should not be used to obtain highly accurate energetic properties.

Page 10 the subsentence "outperform ReaxFF by more than an order of magnitude" should be removed. The NNPs and ReaxFF potentials should not be compared as mentioned in the earlier report.

Page 24 : the subsentence "accuracy close to the reference meta-GGA DFT level" should be reformulated given the errors of the NNP.

Typos : In the abstract "retaining the the reference" should be "retaining the reference"

Reviewer #4 (Remarks to the Author):

The authors did a good job in answering all of my comments (and, as far as I can say, also most of the comments of other 3 reviewers)and hence I recommend the publication of the manuscript in its present form.

RESPONSE TO REVIEWERS' COMMENTS

We thank the Reviewers for their appreciation of the improvements made to the manuscript and for the few minor outstanding suggestions by the Reviewer #3.

We have revised the manuscript following the minor suggestions of the Reviewer #3.

Below, we have responded to each point (in black) made by the reviewers with our comments (blue). In addition, we have provided the manuscript file with the changes explicitly highlighted (olive).

Reviewers' comments:

Reviewer #1 (Remarks to the Author):

I think the authors have done a good job in responding to the questions I raised, and most importantly they now share the training data and models produced, which is in agreement with research data policies.

I think this is a good manuscript, and deserves to be published. It's (in my view) too technical and not novel enough for Nature Communications, however (for the reasons highlighted in my original report and those of the other reviewers).

We thank the reviewer for positive evaluation of the improvements made to the manuscript.

Reviewer #2 (Remarks to the Author):

The authors have improved the manuscript. This could be published as is.

We thank the reviewer for positive evaluation of the improvements made to the manuscript and the recommendation to publish.

Reviewer #3 (Remarks to the Author):

The authors made major efforts to improve the manuscript, which is appreciated. However there remain some concerns as detailed below.

Whereas some statements have been rephrased in the abstract, some sentences are still very general such as “we have developed a general reactive ... including the full range of experimentally ..”. This remains a very bold statement, certainly when put in perspective to some other general purpose machine learning potentials which have recently appeared such as the MACE-MP-0 model, which furthermore seem to have lower errors for the trained energies and forces. The MACE-MO-0 model has MAE of 20meV/atom and 45 meV/Angstrom for the energies and forces (see also some remarks later in this report on the errors). The currently proposed neural network potential certainly has its added value, but it must be situated in a range of similar efforts to train general purpose machine learning potentials. I suggest that the authors add a paragraph to situate the work better in the overall efforts currently taking place in the field.

We thank the reviewer for the positive evaluation of the improvements made to the manuscript and for pointing us to some of the very recent work (<https://arxiv.org/abs/2401.00096>) that was preprinted after the submission of the current work. Indeed, these universal general purpose machine learning potentials (MLPs), which

cover almost the whole periodic table are rather remarkable and have a big potential for screening purposes.

However, as we have mentioned in the Introduction section already in the original manuscript, our approach locates itself in a gap between the universal all-periodic-table MLPs (such as MACE-MP-0, DPA-2, PFP, CHGNet, etc.), and those MLPs that are trained for specific reactive events or thermodynamic state points. In addition, they have significant limitations in describing activated/catalytic events (see, e.g., performance of MACE-MP-0 for catalytic reaction in Figure 2 in <https://arxiv.org/abs/2401.00096> or RMSE force errors for multiple test systems discussed in Supplementary Information reaching hundreds of meV/Å). This is understandable as these models (e.g., MACE-MP-0, CHGNet, GemNet-OC) are typically trained on close-to-equilibrium structures (static geometry optimizations or single points on the experimental crystal structures) without inclusion of any reactive events whatsoever.

It is worth also noting that training on close-to-equilibrium structures means training on a smaller range of force amplitudes, which typically leads to smaller force errors.

In contrast, as we discussed in our previous round of review, our model explicitly targets, besides close-to-equilibrium ambient conditions, reactive events as well as the behavior of the system at elevated temperatures and pressures.

Hence, *the universal all-periodic-table MLP models such as MACE-MP-0 are not directly comparable with our model.*

But we have expanded the existing paragraph discussing the universal general purpose model in the Introduction and added further references to these very recent developments to better highlight this point (pages 3-4).

Regarding the issue of boldness, as addressed in the previous round of the review, we have softened our claims across the whole manuscript, linking it wherever possible to the extensive benchmarking on which we base these claims. Also, as noted above, we do not claim to be general outside this specific class of materials, *just general for this specific class of materials* and support it by extensively testing across multiple conditions, Si/Al ratios, water contents, zeolite frameworks and types of reactive events. Hence, we expect that for this material class, we really are general and reactive and do cover a very broad range of conditions and compositions.

We have removed the word "general" from the abstract to avoid comparisons with the universal general purpose MLPs such as MACE-MP-0. This should avoid any unnecessary or inappropriate comparisons.

It is also suggested to change the sentence : "This NNP combines dramatic sampling acceleration" to "This NNP has the potential to dramatically improve sampling." In the opinion of this reviewer, the sampling can be accelerated as the cost to evaluate the forces is dramatically reduced with a machine learning force field.

The phrase has been changed based on the suggestion of the reviewer.

Page 3 : The sentence " A state-of-the-art tool for accelerating the reactive sampling in" should be changed. ReaxFF can certainly not be regarded as a state of the art tool for simulating reactive events in zeolite chemistry. This point was also made in the earlier review round.

The phrasing has been changed based on the suggestion of the reviewer.

Regarding the temperatures used for training : it is certainly true that using higher temperatures for training is commonly done, however the temperatures of 3600 K seem to be excessively high also compared to other literature references. As it is written now, it is

suggested that simple increase of temperatures is a good methodology to sample more efficiently phase space for activated events, whereas more advanced enhanced sampling methods are generally necessary to efficiently explore phase space. The authors also address partly these issues on page 30, however the authors should comment on this point of sampling so ensure that reader is not misled.

We mention already in the original manuscript both in "Database generation and training of the general BAS zeolite NNPs" and " Dataset generation" sections that we use some biased AIMD data (mostly for Si-O(H) and Al-O(H) bond cleavage mechanisms in H-CHA zeolite) for the dataset generation.

Also, as discussed at length in our previous round of review, we do believe that for our purpose, i.e., generating the robust unbiased database for general potential for a given material class covering a breadth of reactive events under various conditions, use of high temperature AIMD simulation is more efficient than focusing solely on specific a priori expert-chosen reactive events, as it limits the human bias and data redundancy, and samples a lot of distinct and often a priori unknown reactive events with high probability. Also, in the case of (oxide) materials with high melting points such as silica, also other groups have used MD simulations at temperatures in the range of 3000-6000 K to sample activated events in the liquid state (we have added two refs. as examples:

<https://www.nature.com/articles/s41524-020-00367-7>,

<https://www.nature.com/articles/s41524-022-00768-w>), and thus we must disagree that 3600

K is an excessively high temperature for this purpose. Nevertheless, we agree with the Reviewer that supplementing high temperature AIMD by some enhanced sampling data is useful, if one knows what kind of reactive events the MLPs is expected to cover. On the other hand, focusing only on those would provide the developer with the MLP applicable only for the reactive events (or similar) to what it was trained on, i.e., constructing system-specific MLPs trained for a specific (set of) reactions or thermodynamic state points.

Nevertheless, we agree with the reviewer that this point is not sufficiently discussed in the current version of the manuscript, and thus we have highlighted in "Database generation and training of the general BAS zeolite NNPs" section the usefulness of the enhanced sampling methods to obtain relevant data for MLP training, in particular for system-specific MLPs (page 6).

There are some concerns regarding the errors of the energies and forces. The authors mention errors of ca 5 meV/atom and 186 meV/angstrom for the energies and forces respectively, which seem to be substantially larger than errors acceptable for machine learning potentials, currently being devised. On page 6, it is mentioned that these errors are similar to other reactive rotationally invariant MLPs and about the same order of magnitude compared to their own work. With current generation MLPs, errors are normally lower. The authors should comment on this, if the obtained errors are of this scale due to the usage of rotationally invariant network, they should clearly state this. In general terms, it is advised that they add a paragraph mentioning what the acceptable errors are with the current generation MLPs. With the given errors, the NNP will probably be good to obtain a first insight on possible reactive events within zeolite catalysis, but should not be used to obtain highly accurate energetic properties.

We have discussed a similar issue (of rather high training force errors and its effects on accuracy of MLP predictions) in our previous round of review at length with Reviewer #4, to their satisfaction. In summary, the NNPs tend to perform as well (or as poorly) as different "flavors" of XC DFT functionals (e.g., PBE vs. SCAN), as shown in the following tests (further details and tests can be found in our last response to Reviewer #4):

- Comparison of optimized pure silica polymorphs and zeolites (NNP, SCAN+D3(BJ), PBE+D3(BJ)) with experimentally observed (relative) energies, densities, average Si-

- O bond distances, lattice parameters (see Supplementary Table 4) for which the NNPs show same performance as both XC functionals
- Water adsorption energies on all distinct T-sites (T1-T12) in H-MFI zeolite (see Supplementary Figure 7) from NNP, SCAN+D3(BJ) and PBE+D3(BJ) optimizations show that the NNPs deviate as much as PBE+D3(BJ) from the SCAN reference

We also note on page 11 that GGA-level functionals show an RMSE of about 28 meV/atom for formation energies with respect to experiment, a few-fold higher than typical MLP errors. Following the reviewers suggestion, we further comment on the NNP accuracy compared to (meta)GGA accuracy in the manuscript:

- We note on page 7 that the energy/force RMSE of PBE+D3 vs SCAN+D3 (up to 30 meV/atom and 200 meV/Å for high temperature MDs such as silica glass melting) are comparable or higher than the NNP errors
- We added a comment that NNPs with about the same force accuracy were shown to accurately reproduce equilibrium properties such as vibrational density of states (page 11)
- We comment in more detail on the NNP errors of 5-10 kJ/mol for activation barriers (NEB calculations) which are below the average errors of (meta)GGA functionals with respect to coupled cluster calculations (20-30 kJ/mol) on page 19 (see e.g., <https://doi.org/10.1080/00268976.2017.1333644>)

Hence, based on this extensive testing and comparisons, we find the MLP to provide (meta)GGA level quality of description for this material class including possible reactive events - for in-domain systems we expect it to be close to the reference SCAN-D3(BJ) level and for out-of-domain cases to provide energetic (and other) properties with the accuracy similar to some flavor of (meta)GGA XC DFT functional.

Of course, when highly accurate energetic properties beyond (meta)-GGA accuracy (hybrid-DFT, RPA, etc.) are needed, the delta-learning approach discussed and tested at length in the "Improving the baseline model accuracy using delta-learning" is straightforwardly applicable.

Regarding the training errors, based on multiple "head-to-head" comparisons between rotationally invariant and equivariant message-passing networks, the equivariant networks exhibit 2-5 times lower training errors in forces (and 2-3 times lower energy errors, see e.g., <https://www.nature.com/articles/s41467-022-29939-5>). In line with these benchmarks, our preliminary tests showed approx. 3 times lower RMSEs upon switching from a rotationally invariant (SchNet) to a rotationally equivariant model (PAINN) using similar datasets as the one presented in this work.

As previously mentioned, the adoption of the SchNet architecture in the current work is due to historical reasons, as the bulk of the data and validation tests was generated between 1-3 years ago, i.e., when the first works on equivariant MLPs were preprinted. We have since switched to equivariant networks and our future production runs will be or are being done using these architectures. Nevertheless, even the current model is, as discussed above robust and able to retain the (meta)-GGA XC DFT functional quality, and thus has good utility, especially when combined with the delta learning approach described herein.

We have added a description of the errors typically obtained by a rotationally equivariant message-passing networks into the "Database generation and training of the general BAS zeolite NNPs" section, and provided further context about what are the expected consequences of different levels of training errors in practice (page 7).

Page 10 the subsentence "outperform ReaxFF by more than an order of magnitude" should be removed. The NNPs and ReaxFF potentials should not be compared as mentioned in the earlier report.

The phrase has been removed based on the suggestion of the reviewer.

Page 24 : the subsentence “accuracy close to the reference meta-GGA DFT level” should be reformulated given the errors of the NNP.

As we have extensively argued above, and supported by a battery of benchmarks, the current model is truly expected to be close at least to the GGA and/or meta-GGA XC DFT functional level for this particular class of materials.

Nevertheless, we have reformulated the sentence, removing the mention of a specific meta-GGA reference to which we retain the accuracy.

Typos : In the abstract “retaining the the reference” should be “retaining the reference”

The typo has been corrected.

Reviewer #4 (Remarks to the Author):

The authors did a good job in answering all of my comments (and, as far as I can say, also most of the comments of other 3 reviewers)and hence I recommend the publication of the manuscript in its present form.

We thank the reviewer for positive evaluation of the improvements made to the manuscript and the recommendation to publish.

REVIEWERS' COMMENTS

Reviewer #3 (Remarks to the Author):

The authors have addressed the remaining issues properly. The paper can be published as is.

RESPONSE TO REVIEWERS' COMMENTS

We thank again all the Reviewers for their appreciation of the improvements made to the manuscript and the general recommendation to publish.

Below, we have responded to each point (in black) made by the reviewers with our comments (blue).

Reviewers' comments:

Reviewer #3 (Remarks to the Author):

The authors have addressed the remaining issues properly. The paper can be published as is.

We thank the reviewer for positive evaluation of the improvements made to the manuscript and the recommendation to publish.